# Non-volatile rippled-assisted optoelectronic array for all-day motion detection and recognition

Xingchen Pang[1,7], Yang Wang [1,2,3,7] ✉, Yuyan Zhu [1], Zhenhan Zhang[1], Du Xiang [4,5,6] ✉, Xun Ge [2], Haoqi Wu[1], Yongbo Jiang[1], Zizheng Liu[1], Xiaoxian Liu[1], Chunsen Liu [1,4], Weida Hu [2] ✉ & Peng Zhou [1,3,4] ✉

In-sensor processing has the potential to reduce the energy consumption and hardware complexity of motion detection and recognition. However, the state-of-the-art all-in-one array integration technologies with simultaneous broad-band spectrum image capture (sensory), image memory (storage) and image processing (computation) functions are still insufficient. Here, macroscale $(2 \times 2 \, mm^2)$ integration of a rippled-assisted optoelectronic array $(18 \times 18$ pixels) for all-day motion detection and recognition. The rippled-assisted optoelectronic array exhibits remarkable uniformity in the memory window, optically stimulated non-volatile positive and negative photoconductance. Importantly, the array achieves an extensive optical storage dynamic range exceeding $10^6$, and exceptionally high room-temperature mobility up to $406.7 \, cm^2 \, V^{-1} \, s^{-1}$, four times higher than the International Roadmap for Device and Systems 2028 target. Additionally, the spectral range of each rippled-assisted optoelectronic processor covers visible to near-infrared $(405 \, nm–940 \, nm)$, achieving function of motion detection and recognition.

In the era of technological revolution, characterized by intelligence and information technology, motion detection and recognition (MDR) technology has become increasingly important in various applications, including autonomous driving, security monitoring, road traffic enforcement, military defense and intelligent night vision[1,2]. Current state-of-the-art motion detectors based on complementary metal-oxide-semiconductor (CMOS) image sensors only efficiently detect in bright environments difficult to achieve in dark environments, and cannot extract and process motion information due to lack of memory capacity[3,4]. Moreover, converting tremendous data from widely distributed image sensors into digital form for data transmission and storage also leads to time and power wastage[5]. Two-dimension (2D)

materials show great potential due to their unique properties, including local field adjustability, van der Waals contacts, broadband and high photon sensitivity, making them highly suitable for optoelectronic all-in-one applications[6–10]. However, previous work has not integrated the functions of broadband spectrum detection, non-volatile memory and processing functions into a single device to achieve MDR[11–13]. On the other hand, floating gate devices based on multi-layer stacked heterostructures are difficult to achieve large-scale integration of positive and negative optical storage devices due to the manufacturing process complex[14]. The integration of MDR all-in-one technologies can effectively reduce the energy consumption and hardware complexity of motion image information processing.

[1]State Key Laboratory of ASIC and System, School of Microelectronics, Fudan University, Shanghai 200433, China. [2]State Key Laboratory of Infrared Physics, Shanghai Institute of Technical Physics, Chinese Academy of Sciences, Shanghai 200083, China. [3]Shanghai Frontiers Science Research Base of Intelligent Optoelectronics and Perception, Institute of Optoelectronics, Fudan University, Shanghai 200433, China. [4]State Key Laboratory of Integrated Chip and System, Frontier Institute of Chip and System, Fudan University, Shanghai 200433, China. [5]Zhangjiang Fudan International Innovation Center, Fudan University, Shanghai 200433, China. [6]Shanghai Qi Zhi Institute, Shanghai 200232, China. [7]These authors contributed equally: Xingchen Pang, Yang Wang. ✉e-mail: yang_wang@fudan.edu.cn; xiang_du@fudan.edu.cn; wdhu@mail.sitp.ac.cn; pengzhou@fudan.edu.cn

Snakes possess the remarkable unique ability to sense both visible and infrared light using their eyes and pit organs, allowing them to generate motion visual representations of potential predators or prey regardless of light conditions in their surroundings[15]. Taking inspiration from snakes, an approach has been developed for motion imaging and recognition in the dark environment that extends beyond only visible radiation[14]. This type of vision system detects both visible and near infrared radiation can efficiently sense the motion with limited computing resources regardless of the illuminance conditions, which has inspired us to develop integrated all-in-one all-day MDR technology.

Here, we demonstrate a parallel atomically thin $MoS_2$ optoelectronic array (18 × 18 pixels) on a specially treated commercial sr-SiN$_x$ substrate for the all-day MDR, possessing negative and positive optical detection as well as memory and computation capabilities. Notably, the rippled-assisted optoelectronic (RAO) array exhibits remarkable uniformity in both electrical transport and optoelectronic properties, including memory window and optically stimulated non-volatile states. The RAO processor not only enables continuously reconfigurable non-volatile positive and negative photoconductance (NV-PPC and NV-NPC), but also enhances room-temperature mobility (maximum 406.7 cm$^2$ V$^{-1}$ s$^{-1}$) and optical storage dynamic range (over 10$^6$). Moreover, the RAO array mimics the characteristics of a snake vision system, showing a broadband spectrum detection from the visible (405 nm) to the near-infrared (940 nm) range for all-day MDR. Overall, this work presents a high level of integration for motion targets image detection and establishes a model platform for integrating future intelligent optoelectronic devices for versatile engineering applications.

## Results

### Model of the rippled-assisted optoelectronic array

Figure 1a–b demonstrates the 3D schematic and optical image, of the RAO array (18 × 18 pixels) fabricated using a large-area monolayer rippled $MoS_2$ synthesized by chemical vapor deposition (CVD) method. To enhance the optical absorption, a specially treated sr-SiN$_x$ layer is used as the dielectric under the monolayer $MoS_2$ in each processor of the array (see Method for further details on the treatment process of the sr-SiN$_x$ layer). Figure 1c shows the high-resolution scanning transmission electron microscope image, illustrating the RAO processor with the interface of the monolayer $MoS_2$ and the rippled dielectric layer. The energy dispersive X-ray spectroscopy element mapping characterization of the RAO processor is shown in Supplementary Fig. 2. Figure 1d, e present the Raman spectra of the selected 64 RAO processors and the distribution according to the position of the array respectively, validating the uniformity of the as-grown and post-transferred monolayer $MoS_2$ films. The dynamic evolution of non-volatile positive photoconductance (NV-PPC) with a single 10 ms optical pulse is shown as the blue curve in Fig. 1f, and the dynamic evolution of non-volatile negative photoconductance (NV-NPC) with the same optical pulse is shown as the red curve. Figure 1g, h demonstrate the proposed mechanism for the non-volatile photoconductance. The sr-SiN$_x$ dielectric contains abundant hole-trapping centers due to the widely distributed Si-Si bonds[16]. When a large positive gate voltage is applied, the hole-trapping centers can trap injected holes from heavily p-doped silicon. These trapped holes act as a positive local gate, inducing an n-type doping effect and enhancing the electron concentration in the $MoS_2$. The trapping status of the holes maintains even when the gate voltage is removed, which gives a large memory window based on the trap and release of the holes in the dielectric. We conducted long-term measurements of the channel current to assess this memory stability, which shows a remarkable retention ability (See Supplementary Fig. 13). When a small positive gate voltage is applied, the trapping centers in the dielectric are in an empty state. The photon carriers generated by optical stimuli will overcome the $MoS_2$/sr-SiN$_x$ interface barrier and flow along the direction of the electrical field, which brings the trap of the holes in the dielectric according to the direction of the electric field. This leads to an increase of positive charge in the dielectric, resulting in NV-PPC (Fig. 1g). For light erase process, a fixed −3 V gate voltage is applied accompanied by the optical stimulus, in which the holes in the dielectric are released. The reduction of stored holes leads to a smaller channel current, resulting in the negative photo conductance (NPC). The carrier distribution and band diagram of NV-PPC and NV-NPC is shown in Supplementary Fig. 12. We compared the electrical transport properties of the devices with specially treated sr-SiN$_x$ and normal SiN$_x$, with different roughness (See Supplementary Fig. 4). The wet polishing method is used to decrease the roughness of the substrate, and a more detailed demonstration is shown in Supplementary Fig. 5. The transfer curve of the RAO processor exhibits a large memory window in the anticlockwise direction. The sr-SiN$_x$ dielectric contains rich Si-Si bonds that are widely distributed, leading to abundant hole-trapping centers, which serve as the foundation of the device's memory window. It is noteworthy that the roughness of the dielectric surface can have a significant impact on the optical storage capacity and the dynamic range of the device. The average roughness of the RAO processor is around 3.4 nm (See Supplementary Fig. 4c), which is intentionally created to improve the carrier mobility and the photon current[17]. The RAO processor with rougher surfaces shows much stronger NV photocurrents in comparison to that on smoother surfaces, mainly due to the enhanced mobility with the rippled structure (see Supplementary Fig. 3 for the comparison of the photocurrent and dynamic range of different roughness). Besides, a first-principles calculation clearly demonstrates that both the in-plane and out-of-plane dielectric constants of $MoS_2$ exhibit an increase with the height of curvature. Consequently, the ripples in the $MoS_2$ caused by bulged substrate results in an increase in the dielectric constant, This, in turn, contributes to the enhancement of mobility. (detailed discussion is shown in Supplementary Section 2).

### Electrical transport and optoelectronic properties of the RAO processor

Figure 2a presents the structure diagram of the RAO processor prepared with a $MoS_2$ thin film onto a specially treated sr-SiN$_x$ dielectric. Figure 2b demonstrates the transfer characteristics of the RAO processor sweeping at different gate voltages, where the memory window monotonously increases with increasing gate voltages. When a scanning voltage of 25 V is applied to the device, the memory window expands to 16 V, indicating a strong memory effect induced by the hole-trapping centers. Figure 2c displays the photocurrent curves of the NV-PPC and the NV-NPC showing multiple states under 5 ms periodic optical pulses. Additionally, we conducted thorough extensive testing on the response time of the RAO device with light stimuli lasting from 1 to 50 ms (see Supplementary Information Fig. 9a–f). Remarkably, the RAO device exhibits proper functionality even under light pulses as short as 1 ms. Furthermore, the device continues to perform effectively when exposed to light pulses lasting up to 50 ms. We also extracted the non-volatile photocurrent between the PPC and NPC at different optical pulse numbers, as shown in Fig. 2d. The R-square value for the linear fitting of the non-volatile photocurrent is 0.9995 and 0.99658 for NV-PPC and NV-NPC respectively, indicating the remarkable linearity of the responsivity under continuous optical pulses. Furthermore, we examined the precise conductance extracted from the evolution of the current under periodic optical stimuli (Fig. 2e–f). Both NV-PPC and NV-NPC could be divided into 28 discrete states, indicating their optical programmability for intelligent recognition training in the implementation of convolutional neural networks (CNN) and image recognition. We also investigated the reconfigurable properties of both NV-PPC and NV-NPC. The current curves under continuous periodic optical stimuli with periodically tuned gate

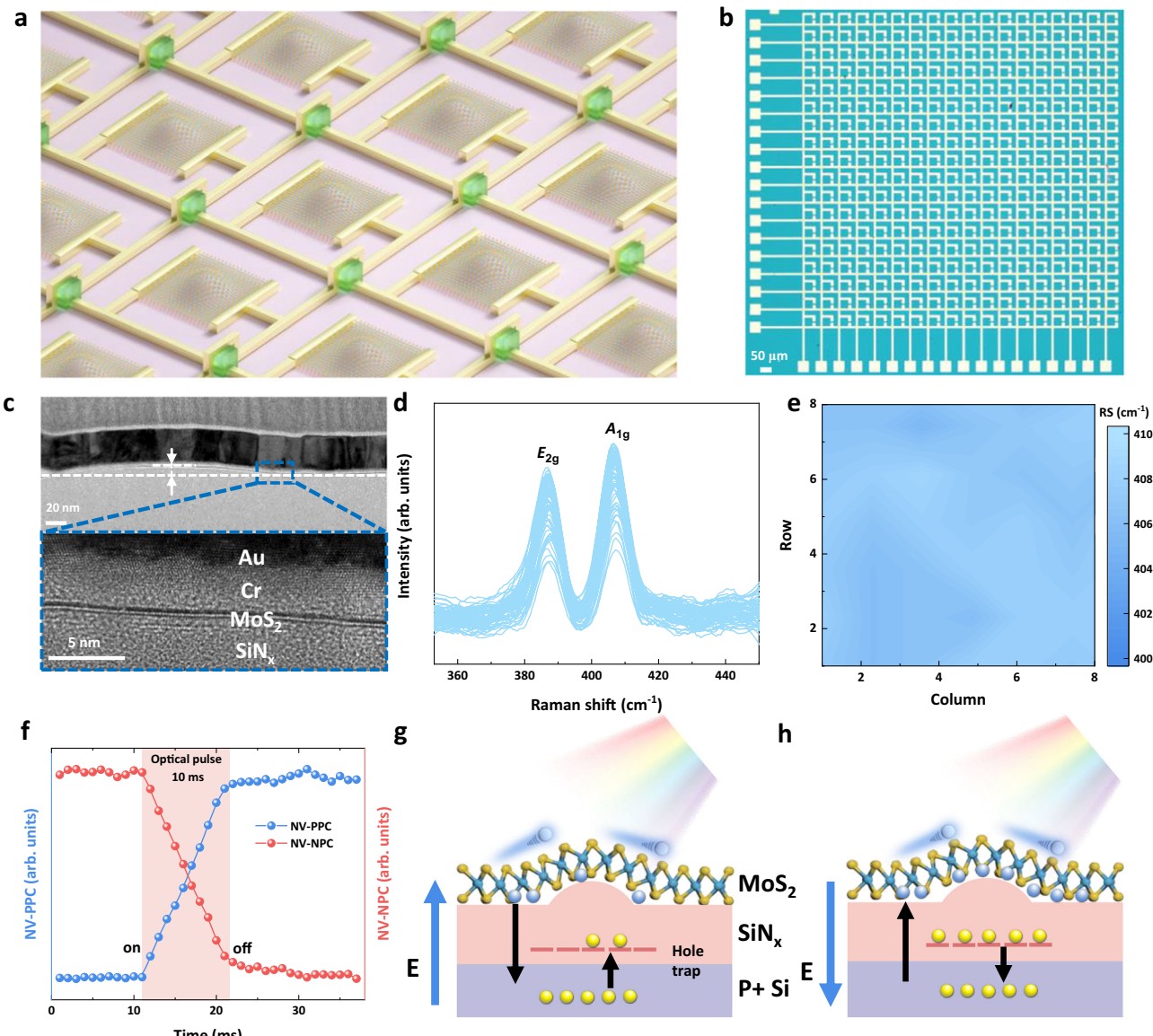

**Fig. 1 | Model of the rippled-assisted optoelectronic array. a** 3D schematic of the RAO array, which shows a simple structure of each pixel. **b** Optical image of the RAO array. **c** Scanning transmission electron microscopic characterization, which shows the rippled interface and the monolayer $MoS_2$. **d** Raman spectra of the selected 64 MDR phototransistors. **e** Color mapping of the Raman peak of the selected 64 MDR phototransistors according to the position. **f** Dynamic evolution of NV-PPC and NV-NPC with a single 10 ms optical pulse. The orange region represents the duration of the optical pulse. **g**, **h** Mechanism of NV-PPC and NV-NPC with enhanced electron mobility. The yellow and blue spheres represent the hole and electron, respectively. The underline represents the natural hole-trapping center. Source data are provided as a Source Data file.

voltage were measured, which determines the sign of the photocurrent, with the repetition of 50 cycles, as shown in Supplementary Fig. 20. The NV-PPC and NV-NPC in the cyclic test are extracted and demonstrated in Fig. 2g–h, where the conductance at each state is observed in a certain range without significant outlier, indicating the outstanding cyclic endurance of the RAO processor. It is also noted that the neighboring states are almost completely separated with negligible overlaps, further validating the excellent stability of the NV-PPC and NV-NPC. Furthermore, we investigated the relationship between the photocurrent and drain voltage ($V_{DS}$) (see Supplementary Fig 15), where both the NV-PPC and NV-NPC are linearly modulated by changing the $V_{DS}$ from 0.1 V to 0.9 V, indicating the great potential for the implementation of CNN. To ensure the continuous operation of the RAO processor, we carried out the optoelectronic measurements under a wide range of laser powers and pulse numbers at different wavelengths, with the extraction of all the NV-PPC and NV-NPC from

the current curves. The measurements in Fig. 2c–h are all experimented with laser pulses at 520 nm and other wavelengths are shown in Supplementary Fig 16. Both the NV-PPC and NV-NPC at different wavelengths are plotted in 2D color mapping. As the pulse number and the laser power increase, the absolute value of photocurrent demonstrates the consistent enhancement trend for all wavelengths (Supplementary Fig 17–19), leading to similar color mappings. Furthermore, widening the detection wavelength of the optoelectronic processor to the near-infrared band (~940 nm, Fig. 2o–p), illustrating its broadband photoresponse and artificial hyper-vision functionality. To further verify the functionality of the specially treated sr-$SiN_x$, we've tested the property of $MoTe_2$ to better evaluate the enhancement induced by the rippled interface. The rippled $MoTe_2$ device demonstrates the same continuously reconfigurable NV-PPC and NV-NPC, illustrating the rippled sr-$SiN_x$ dielectric layer has good generality and flexibility for providing functionality of both positive

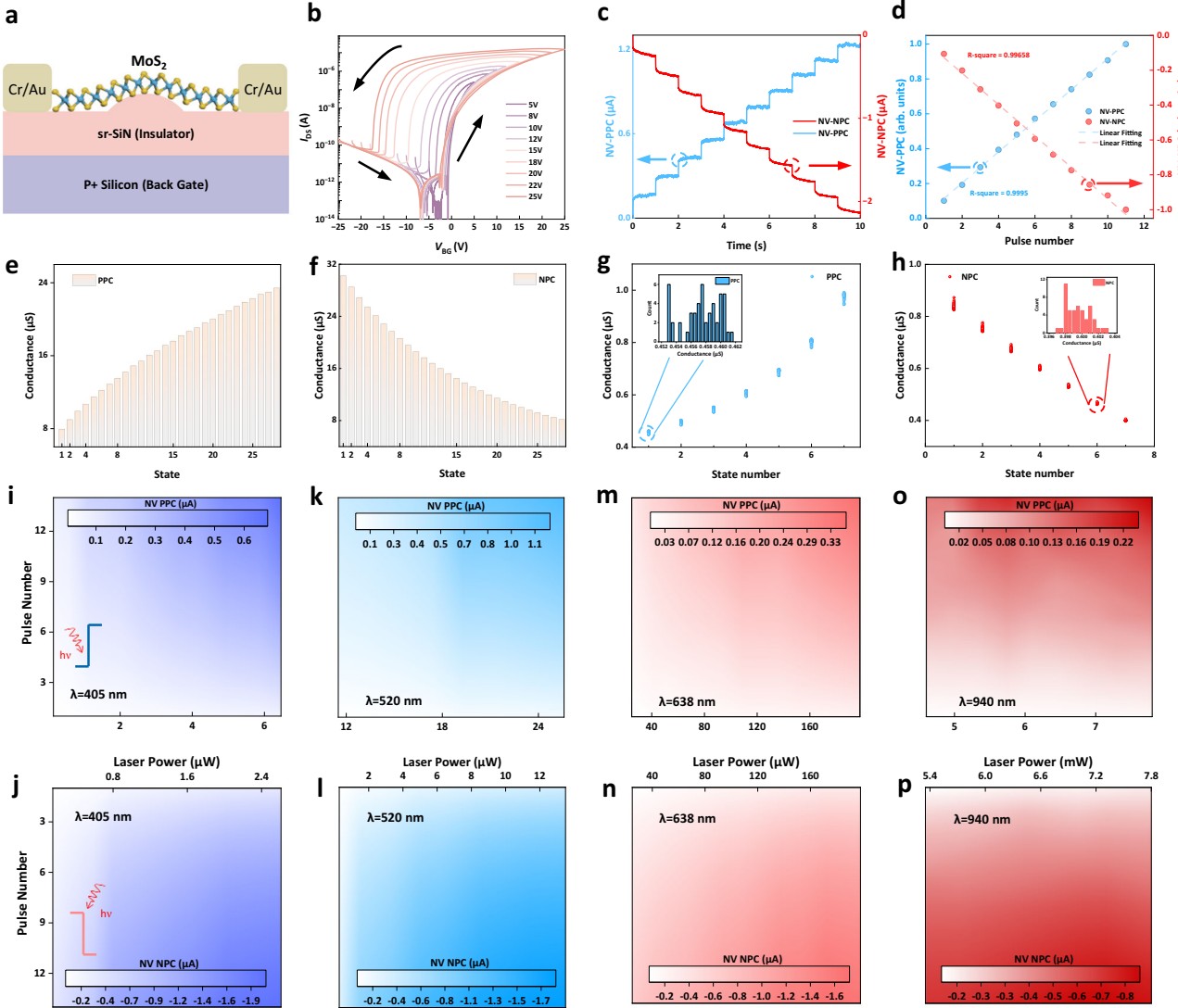

**Fig. 2 | Structure and photoconductivity of the RAO processor. a** Structure of the RAO processor. **b** Transfer curves evolution with changing the ranges of the sweeping gate voltages at $V_{DS} = 0.5\,V$, which indicates a remarkable memory effect of the RAO processor. $I_{DS}$ and $V_{GS}$ denote the drain current and gate voltage respectively. **c** Cumulative positive and negative photoconductivity with progressive multilevel states under periodic 5 ms optical pulses both with 1 s intervals. The red curve represents the NV-negative photocurrent and the blue curve represents the NV-positive photocurrent. **d** Extracted cumulative NV-photocurrent and the linear fitting. The R-square values are 0.9995 and 0.99658 for NV-PPC and NV-

NPC, respectively. **e, f** 28 discrete conductance states of both NV-PPC and NV-NPC excited by the same optical pulse, which shows the accurate programmability for weight training. **g, h** Conductance at each state under continuously periodic light stimuli for stabilization test of both NV-NPC and NV-PPC. The distribution of a certain state from 50 periods is shown in the insets. **i–p** Colour mapping of both cumulative NV-PPC and NV-NPC. **i, k, m, o** represent the NV-PPC and (**j, l, n, p**) represent the NV-NPC, as (**i, j**) with 405 nm laser, (**k, l**) with 520 nm laser, (**m, n**) with 638 nm laser and (**o, p**) with 940 nm laser. Source data are provided as a Source Data file.

and negative photo response and memory capacity for some types of Transition Metal Dichalcogenides (See Supplementary Section 10 and Supplementary Figs. 25–31).

## Uniformity in electrical and optical statistics of the RAO array

The RAO processor features a simple CMOS-compatible architecture and possesses outstanding performance in integrated sensing, storage and processing, which demonstrates great potential for scalable array fabrication enabling efficient motion detection and recognition. We performed a comprehensive assessment of MoS₂ characteristics obtained through both chemical vapor deposition (CVD) synthesis and mechanical exfoliation (ME). The comparison between devices fabricated from CVD and ME films, featuring identical channel length and width, holds significant value in elucidating the consistency and performance of these materials in electronic applications (shown in Supplementary Figs. 22–24). We characterized the electrical transport and

optoelectronic properties of an RAO array of 324 devices designed for MDR to obtain significant figure-of-merits. As shown in Fig. 3a, 262 devices demonstrate consistent transport characteristics without significant fluctuations in dark conditions, suggesting a high yield of 80%. An apparent gap exists for the transfer curves sweeping in different directions, which illustrates the uniformity of the memory window. Furthermore, we performed a statistical analysis of the carrier mobility ($\mu = \frac{dI_{DS}}{dV_{GS}} \times [L/WC_iV_{DS}]$, where $L$, $W$ and $C_i$ are the channel length, the channel width, and the capacitance between the channel and the back gate per unit area, respectively) from the transfer curves, of which the statistical data shows a similarity to normal distribution. The mobility statistics of the RAO array illustrate a normal distribution as shown in Fig. 3c, illustrating that the specially treated sr-SiNₓ dielectric layer significantly enhances carrier mobility, with a maximum value of 406.7 cm² V⁻¹ s⁻¹. This is notably higher than the mobility observed in the conventional transistors made with monolayer MoS₂ synthesized

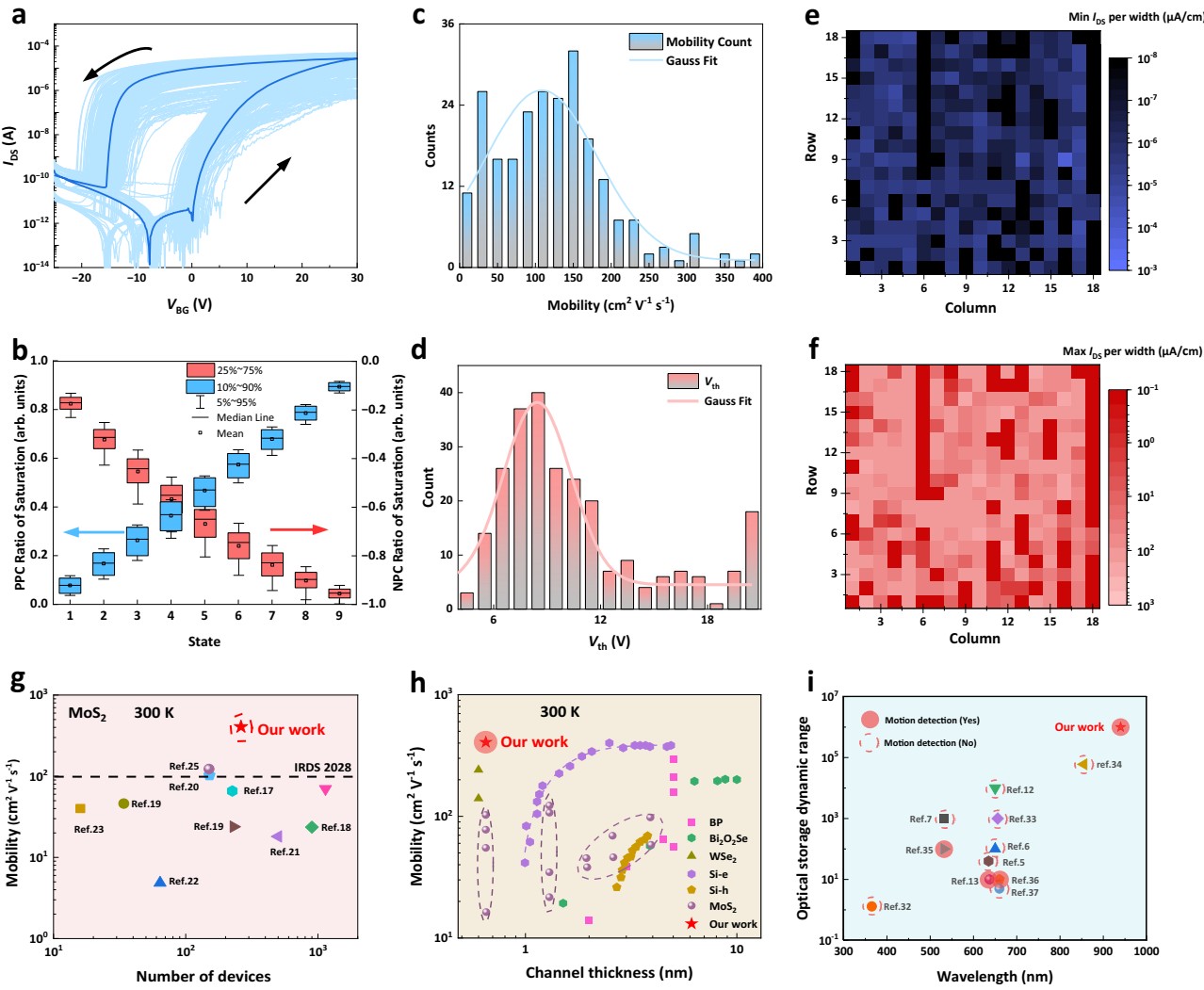

**Fig. 3 | Electrical and optical statistics of the RAO array. a** Transfer curves of 262 pixels with a high yield of 80% at the $V_{DS} = 0.5$ V. The deep blue curve represents a typical transfer curve of each RAO processor, which shows a large memory window. **b** State statistics induced by optical stimuli for both NV-PPC and NV-NPC. A slight overlapping is observed between the two adjacent states, indicating outstanding uniformity. **c** Mobility statistics of the RAO array illustrate a normal distribution. The maximum and average mobilities are 406.7 and 116.5 cm² V⁻¹ s⁻¹. **d** Threshold voltage statistics of the RAO array exhibit a similar normal distribution as that of the mobility. Both (**c**, **d**) prove the remarkable uniformity of the RAO array. **e** Minimum $I_{DS}$ per width of each pixel according to its position. **f** Maximum $I_{DS}$ per width of each pixel according to its position. **g** Benchmark of electron mobility of MoS₂

arrays synthesized by CVD, where the maximum electron mobility of our devices is 406.7 cm² V⁻¹ s⁻¹ exceeding the target of the IRDS 2028. **h** Statistics of electron mobility versus channel thickness. The mobility of the RAO processor is superior to bulk silicon while the channel thickness is reaching the atomic level. **i** Comparison of optical dynamic range under different wavelengths for intelligent image perception technology in comparison to the previous works. The solid boxes represent intelligent image perception techniques that can be used for moving target detection. The RAO array achieves the MDR functions with the largest dynamic range and wavelength band among the previous works. Source data are provided as a Source Data file.

by the CVD method. Additionally, we extracted the threshold voltages of all the phototransistors using the constant-current method. The arbitrary constant drain current is set by the equation $I_{constant} = (W_m/L_m) \times 10^{-7}$, where $W_m$ and $L_m$ are the mask channel width and length respectively, and the constant current is 0.2 μA[18]. The statistics of the threshold voltages were similar to a normal distribution and parallel to the distribution of the carrier mobility (Fig. 3d). Most of the threshold voltages of the phototransistors fall within 6−12 V. Furthermore, the uniformity of the maximum and minimum current of the transfer curves, where the majority of the currents are within a range of one decade are shown in the color mapping plot in Fig. 3e–f. The minimum source-to-drain currents per width of most phototransistors are below $10^{-5}$ μA/cm at the $V_{DS} = 0.5$ V, which approaches the limit of the test instrument. On the other hand, the maximum source-to-drain currents typically exceed $10^2$ μA/cm.

Figure 3b illustrates the data of both the NV-PPC and NV-NPC for the RAO array under periodic and continuous optical pulses, presented in a box chart, displaying the distribution, mean and median for each state. Despite the large amounts of pixels, the relatively low overlap between each pair of adjacent states indicates excellent stability and functionality, making the RAO array suitable for MDR. The RAO processor also exhibits outstanding electrical transport performance in contrast to other 2D semiconductors and conventional silicon CMOS devices. Notably, our transistors exhibit a significant advantage in carrier mobility, as shown in Fig. 3g–h, surpassing that of all previous works within the range of transistors based on CVD MoS₂, and exceeding the IRDS 2028 target[19–32]. It is worth mentioning that the RAO array demonstrates the highest mobility within the 2D family as the channel thickness approaches the limit of a single atomic layer. To discern the impact of contact resistance and channel resistance, we

utilized a Transmission Line Model to determine the specific contact resistivity of the metal-semiconductor junction (shown in Supplementary Section 11). The comparison of optical dynamic range under different wavelengths for intelligent image perception technology among previous works is shown in Fig. 3i[5–7,13,14,33–38]. The RAO array has reached both the highest dynamic range (over $10^6$) and the broadest band spectra for the MDR functions in comparison to the previous works.

## Broadband spectrum for all-day MDR functions

Snakes provide an ideal model for understanding the hyper-vision function as their eyes are capable of detecting visible light while their pit organs can detect infrared radiation. This dual sensory mechanism allows snakes to create a "thermal image" of moving predators or prey. By combining thermal and visual information in their brains, snakes exhibit exceptional precision in both bright and dark environments. Figure 4a depicts how a snake's sensory system detects both visible and infrared radiation under different light circumstances. As an example, the snake can detect motion prey such as a mouse through optical radiation in bright environments, whereas in dark environments, the pit organ enables them to sense infrared radiation to locate motion prey. Figure 4b shows the working principle of all-day moving target detection and recognition. The MDR functions can be achieved by combining two arrays with opposite signs of photoconductivity. Each motion process can be conceptualized as an image stream at different moments. For instance, the array with positive photoconductivity detects the objective in frame n, while the objective in frame n + 1 is detected by the array with negative photoconductivity. Each RAO processor in either array outputs an NV-photocurrent that is proportional to the photoconductivity multiplied by the input laser power. When combining the photocurrent of the two arrays, the photocurrent of pixels receiving static objective with photoconductivity of opposite signs cancels out. Since the magnitudes of the photoconductivity for both positive and negative photoresponses are similar, the sum of the photocurrents is close to zero, which makes it easy to obtain the resulting frame of only the motion information of the objective. If there's no motion during alone the frames, most of the pixels are dark because the photocurrent with opposite signs cancels each other out. As demonstrated in Fig. 4c, the RAO array is capable of detecting the motion of a person riding an electric bicycle in a bright environment through the radiation of the visible or broadband spectrum. The detection results obtained through different spectra are similar. Figure 4d displays the distribution of the brightness output across the two spectrum bands, indicating that motion detection can be accomplished in a bright environment using only the visible spectrum. However, in dark environments, using only the visible spectrum is insufficient. The original and detected results obtained from both the visible and broadband spectra are shown in Fig. 4e. Since the visible part of the input is obscure, the comparison through different spectrum bands is more evident. With the assistance of near-infrared radiation, the RAO array is capable of detecting motion even in a dark environment. Additionally, the brightness detection results for two spectrum bands in the dark environment are displayed in Fig. 4f. The distribution of the detection result using broadband spectrum shifts right compared to the visible spectrum, indicating that the broadband RAO array can achieve the all-day MDR functions.

## Discussion

We developed a subtle approach to achieve the all-day MDR, based on a parallel atomically thin $MoS_2$ optoelectronic array (18 × 18 pixels) on a specially treated commercial sr-$SiN_x$ substrate. The RAO processor not only enables continuously reconfigurable NV-PPC and NV-NPC, but also significantly enhances room-temperature mobility and optical storage dynamic range. Moreover, the RAO array exhibits remarkable uniformity in both electrical transport and optoelectronic properties,

including the memory window and optically stimulated non-volatile states. Each RAO processor combines the functions that take inspiration from the snake's eye and pit organs, enabling the array to detect moving targets in both bright and dark environments. The rippled $MoTe_2$ processor also demonstrates similar continuously reconfigurable non-volatile positive and negative photoconductance, illustrating the consistent optoelectronic properties of the rippled structure. This work provides a simple technique to achieve a high level of integration for all-day MDR functions, and establishes a model platform for integrating future intelligent optoelectronic devices for the artificial hyper-vision bionic and autonomous driving design applications.

## Methods
### Device fabrication

As for the device fabricated using mechanically exfoliated crystals, the bulk $MoS_2$ samples were purchased from HQ Graphene. The $MoS_2$ flakes were exfoliated and transferred onto the highly p-doped silicon substrate with 500 nm sr-$SiN_x$ on top by a fixed-point transfer technology. The sr-$SiN_x$ is manufactured by Silicon Valley Microelectronics, and was grown by the standard low-pressure chemical vapor deposition (LPCVD) technique. The roughness of the sr-$SiN_x$ layer can be tuned with pressure and the reactant gas flow during the LPCVD growth process. Standard electron-beam photolithography (EBL) was used to fabricate the source (S) and drain (D) electrode patterns. Afterward, Cr/Au (5/30 nm) electrodes were deposited by electron-beam evaporation (EBE). After the lift-off process, the $MoS_2$ all-in-one phototransistor was successfully fabricated. For the RAO array, the monolayer $MoS_2$ sample supplied by Six Carbon Shenzhen was synthesized by CVD technique on the sapphire substrate and then transferred onto the $SiO_2$ substrate. The specific process flow for large area growth $MoS_2$ was obtained from Six Carbon Shenzhen. $MoO_3$ (purity over 99.999%) is used as the Mo source and solid sulfur is used as the source (s). The process is taken in the Ar atmosphere in the tube furnace of which the diameter is 80 mm with a double temperature zone. $MoO_3$ is heated to 650 °C and solid sulfur is heated to 180 °C with pressure 4000 Pa in the tube. The process takes 10 min and the $MoS_2$ is synthesized on the sapphire surface. Then the monolayer $MoS_2$ will be carefully transferred into a target chip like Si/$SiO_2$ substrate. After that, the $MoS_2$ on the $SiO_2$ substrate was spin-coated with polystyrene (PS) and kept at 90 °C for 10 min. Then both the monolayer film and the PS were separated from the $SiO_2$ substrate using deionized water, and transferred onto a polydimethylsiloxane (PDMS). After that, the PDMS substrate carrying the $MoS_2$ film was transferred onto the specially treated sr-$SiN_x$ substrate by the fixed-point transfer technology keeping 100 °C within the transfer process. The substrate, monolayer $MoS_2$ and the PS layer were then soaked into the methylbenzene to remove the polystyrene (PS) layer. Source electrodes were patterned using laser direct writing (LDW) technology (Micro-Writer ML3) and subsequently deposited by EBE. The separation layer ($SiO_2$) at the junction of the source and drain electrodes was patterned using LDW technology and deposited by physical vapor deposition (PVD, Lesker PVD 75). Then, drain electrodes were patterned using LDW technology and deposited by EBE. The $MoS_2$-channel region was obtained by $O_2$ plasma etching. After the lift-off process, the RAO array was successfully fabricated.

### Characterizations and measurements

The surface morphology of normal silicon nitride and the specially treated sr-$SiN_x$ were characterized using Atomic Force Microscopy (AFM). To examine the number of layers of the $MoS_2$ synthesized by the CVD method, a cross-sectional analysis was performed using high-resolution transmission electron microscopy (HRTEM) technology with energy-dispersive X-ray spectroscopy (EDS) elements mapping analysis. The $MoS_2$ material was also characterized using Raman spectroscopy, which revealed strong peaks near 388 $cm^{-1}$ and

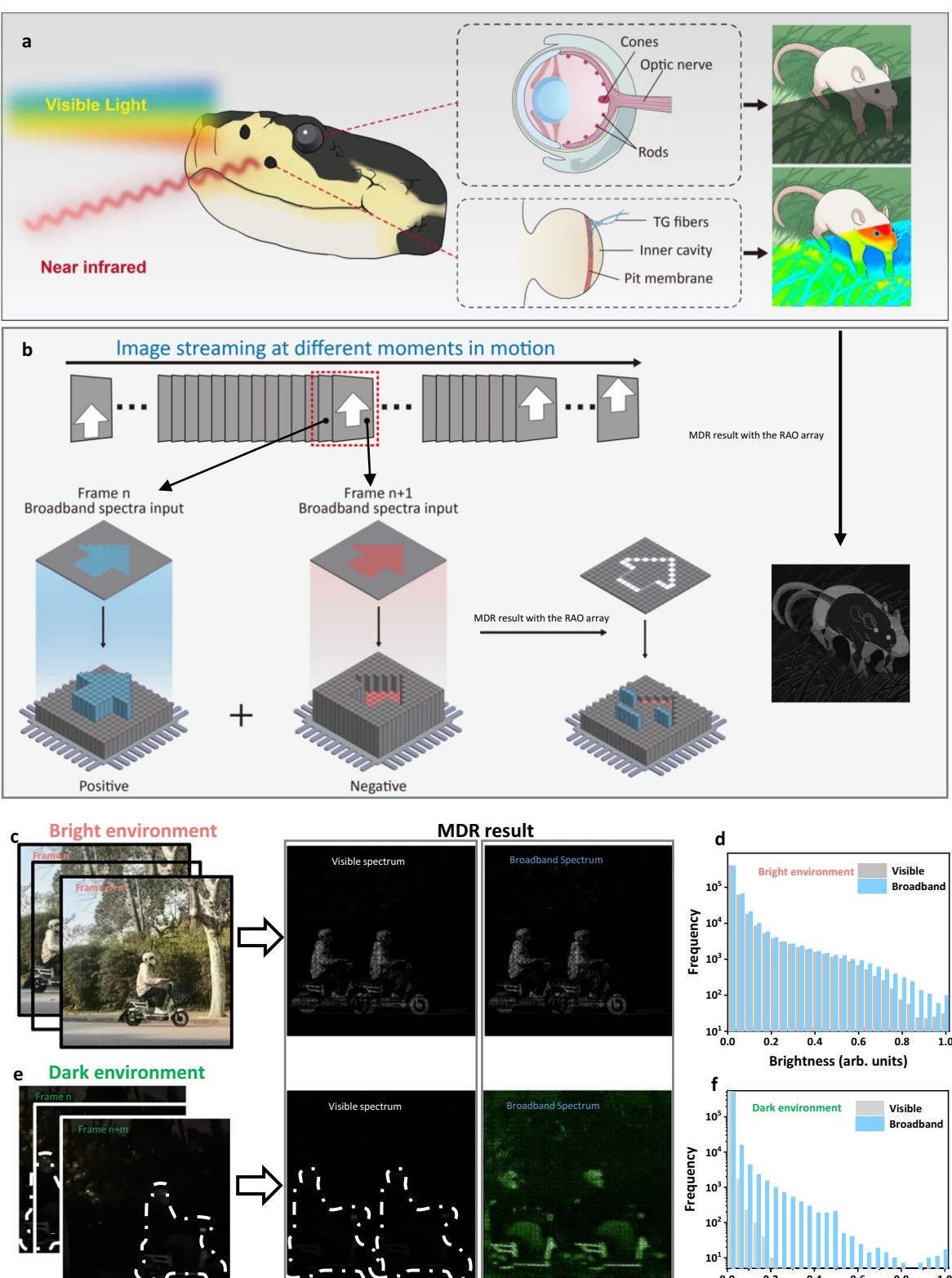

**Fig. 4 | Mixed spectra for all-day motion detection and recognition inspired by snakes. a**, **b** Biological system of snakes for the detection of both visible and infrared radiation, as well as the all-day MDR methodology inspired by the snakes. The array of positive photoconductivity and negative photoconductivity multiply the frame n and frame n + 1, respectively. The sum of the two array makes the static part of the frame cancels off, which allows for motion detection. **c** Motion detection in a bright environment. The results are similar for visible and broadband spectrums. **d** Distribution of the brightness of detection results in a bright environment for both visible and broadband spectrum. **e** Motion detection in dark environment. **f** Distribution of the brightness of detection results in dark environment for both visible and broadband spectrum. Both (**e**, **f**) show a remarkable enhancement in the broadband spectrum due to the extra photoresponse of the near-infrared band. Source data are provided as a Source Data file.

406 cm⁻¹. The electrical transport properties of the RAO array were characterized by the Keysight B1500A semiconductor analyzer. The photoelectric properties were obtained by optical stimulation of lasers with wavelengths of 405 nm, 520 nm, 638 nm and 940 nm (Thorlabs), and the corresponding laser powers were measured using a light intensity meter (Thorlabs).

### Detection and recognition of a person riding an electrical bicycle

The process of implementing motion detection involved several steps. Firstly, the weight matrices of the NPC and PPC were loaded, and each frame of motion was extracted from the input. Next, these frames were divided into smaller frames measuring $18 \times 18$ pixels. Each of these small frames was mapped with the NPC and PPC weight matrices separately. The mapping results of the two matrices were then summed up, and the data was converted into image patterns using MATLAB. In order to simulate real-life motion detection under various lighting conditions, the specific pixel value was extracted from a moving individual riding an electric bicycle in both bright and dark environments.

## Data availability

The data supporting the findings of this study are available within the article and its supplementary files. Any additional requests for information can be directed to, and will be fulfilled by, the corresponding authors. Source data are provided with this paper.

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

## Acknowledgements

This work was supported financially by the National Key Research and Development Program of China (Grant Nos. 2023YFB3611400, W.H.; 2021YFA1200500, P.Z.), National Natural Science Foundation of China (Grant Nos. 61925402, P.Z.; 62090032, P.Z.; 62304040, Y.W.; 62104041, D.X.; and 62374038, D.X.), Shanghai Sailing Program (Grant No. 21YF1402600, D.X.), Science and Technology Commission of Shanghai Municipality (Grant No. 19JC1416600, P.Z.), China National Postdoctoral

Program for Innovative Talents (Grant No. BX20220082, Y.W.), China Postdoctoral Science Foundation (Grant No. 2022M720750, Y.W.), and Open Fund of State Key Laboratory of Infrared Physics (Grant No. SITP-NLIST-ZD-2023-01, Y.W.).

## Author contributions

Y.W., D.X., W.H., and P.Z. conceived the idea and supervised the work. Y.W., D.X., and X.P. designed the experiments. X.P., Y.W., D.X., Z.Z., and C.L. provided assistance with mechanism analysis and discussion. Y.Z. and X.G. analyzed the physical model and provided first principles calculation. X.P. performed device fabrication and characterization. H.W. and Y.J. support the characterization of materials. Z.L. and X.L. provided assistance with the fabrication. X.P. and Y.W. co-wrote the manuscript and all authors contributed to the revision of the manuscript.

## Competing interests

The authors declare no competing interests.
