## [Peer Review File · Nature Communications]

REVIEWER COMMENTS

Reviewer #1 (Remarks to the Author):

General Comments:

“The manuscript describes a novel Van der Waals all-day motion detector, which fully demonstrates the unique properties of Van der Waals materials. The work is very interesting and will further promote the application of Van der Waals materials for motion detection with simple structures and high performance. The results are convincing in the sense that they provide sufficient evidence that rippled-assisted phototransistors are capable of non-volatile memory, broad-spectrum image capture and image processing functions. The rippled interface of silicon nitride and MoS₂ monolayer used in the research have been characterized through various characterization methods, which makes the whole results much more solid. Meanwhile, broadband optoelectronic devices possess good performance with a simple structure, which derives great potential for integrated applications. Among the previously proposed works, the effort has been made to fabricate the light sensor with tunable responsivity using Van der Waals layered materials, which aims to achieve in-sensor-computing functionality. However, the complex structure with multiple layers of different materials limits the yield and thus the application. So far, the method to manufacture optical sensors with sensing, memory and processing functions on a large scale has not been reported. Pang et al in this paper, for the first time, realized all-in-one optoelectronic devices on a large scale through a simple structure consisting of only a monolayer MoS₂ and a layer of silicon-rich silicon nitride. The authors provided sufficient and supportive data to demonstrate the all-day motion detector utilizing the broadband and the rippled-assisted optoelectronic devices, of which the simple structure shows great potential to overcome the limitations of yield and integration. The work will further promote the development of electronics and motion detectors by utilizing the extraordinary properties of Van der Waals semiconductors and the rippled-assisted interface. Hence, I recommend the paper for publication in Nature Communications if the authors will address the following questions:”

Comment 1:

“Response time is a key parameter indicating whether the device is suitable for continuously working in real circumstances. The author should provide the response time data with incident light with different wavelengths.”

Comment 2:

“The authors should provide the responsivity data and its incident power dependence in visible and in infrared.”

Comment 3:

“For visible detectors, the data about electrical and optoelectronic properties of MoS₂, such as output curves are deficient. Please provide detailed data.”

Comment 4:

“The detectors exhibit a quite low dark current. When concerning the area of the devices, the author should change to the current density. It will be more reasonable.”

Comment 5:

“As for the memory capability, the data illustrating the maintenance of the memory property is deficient. Please provide detailed data.”

Comment 6:

“What’s the difference between rippled interface and a flat interface? What enhancement of mobility does the rippled interface bring quantitatively?”

Comment 7:

“In Fig. 2b, is the transfer curve clockwise or anti-clockwise? The direction of the transfer curve to some extent demonstrates the memory mechanism of this device. The author should mark the direction of the transfer curve.”

Comment 8:

The authors also mentioned the dynamic range of their devices. Can they compare with existing works (e.g., Nature Electronics, 2022, 5, 84-91)?

Reviewer #2 (Remarks to the Author):

In this study, the authors show a 2×2 mm² integration of what they claim a rippled-assisted optoelectronic (RAO) array (18×18 pixels) to mimic the characteristics of snake vision systems for all-day motion detection and recognition. The claim of a snake vision almost reads like a buzzword claim. Detection of light of 940 nm is nothing new. In fact MoS₂ has been shown to possess both positive and negative PPC in even broader wavelength ranges than what is reported here using other substrates such as p doped Ge. There are several claims in the paper that are not evidenced properly. Nor was I able to really identify what the key advance here is.

There are multiple issues that need to be addressed:

1. Why were measurements done on mechanically exfoliated MoS₂ when the authors have access to 18 by 18 pixels of CVD grown films?
2. There is no evidence shown to support the claim of 18 by 18 pixels and such large area growth.
3. In the supporting information, suddenly MoTe₂ appears out of nowhere. Is there a rationale for that?
4. The authors claim to have achieved very high room temp mobilities. No information is provided how these were calculated and what mechanism allows such high mobilities

5. The explanation of positive and negative photocurrent is very shallow and highly speculative without any evidence or theoretical modeling. If the explanation provided is to be taken at face value, then every material with a direct bandgap and a Schottky barrier with the substrate will show a similar behaviour which raises questions about the veracity of the explanation provided

Overall, the study is very premature, shows some results but the claims do not really stack up in terms of existing literature in terms of advancement shown. Moreover, key claims are not backed by solid evidence and has a lot of hyperbole in the abstract and introduction. I would suggest that the authors look at other papers on MoS₂ that have shown better photodetection spectral range while showing positive and negative PPC.

Unfortunately, I do not think that this paper nevertheless reports an advance that merits publication in this journal and cannot therefore recommend publication.

Reviewer #3 (Remarks to the Author):

The manuscript titled "Non-volatile Rippled-Assisted Optoelectronic Array for All-day Motion Detection and Recognition" explores a novel approach to motion detection and recognition (MDR) by integrating a rippled-assisted optoelectronic (RAO) array. The study addresses the challenges associated with conventional CMOS image sensors and presents a unique device capable of continuous reconfigurable non-volatile positive and negative photoconductance, enhanced mobility, and a wide optical storage dynamic range. The RAO array aims to achieve MDR in various lighting conditions, including dark environments. Overall it is a good paper and worthy of publication after several comments below are addressed.

Comments:

1. I think the major result is high mobility, not necessarily all the motion detection and recognition. I would suggest the authors do a much more thorough assessment of the mobility and provide standard deviation.
2. The other thing I would request the author to do is checking the mobility on control samples without ripples and provide averages and standard deviations.
3. Finally, the mobility should also be checked as a function of channel length to isolate effect of the contact resistance vs channel resistance.
4. The photo detection and motion detection data is great, I don't have much comments about it. I only want the authors to be clear and comment if the image capture and motion detection is done by the

whole 2D array or just by a single pixel and then rest is simulation. It seems to me that the pixels are not individually wired to sense. So the perception of image and motion capture appears misleading.

Reviewer #4 (Remarks to the Author):

In this work, the authors report some very interesting results on the smart sensing/imaging by combining spectral sensing, storage and simple computation functions all on chip. In my view, optical sensing and imaging are at the verge of substantial transformation. This work may represent one of such changes. Overall, the results are interesting and sound. I think it should be published in Nature Communications after they address the following issue.

It seems that this so-called "Ripple assisted optoelectronic (ROC)" devices play an important role. I suggest that the authors clarify its role. For example, will this demonstration still valid if regular devices with no ripples are utilized? If yes, what is expected performance degradation with regular devices. Also what are the mechanisms for the performance improvement in ROC? All these issues should be thoroughly discussed.

Responses to Reviewers' Comments

We acknowledge the reviewers for carefully reading our manuscript '*Non-volatile rippled-assisted optoelectronic array for all-day motion detection and recognition*' (NCOMMS-23-28827-T) and providing constructive comments on our work. According to the reviewers' comments, we have carefully revised our manuscript and provided more detailed data to improve the manuscript's readability. We reorganized the manuscript and Supplementary Information to present the new data with elaborate discussions and abundant experiment data. With the help of the reviewers, we believe that the revision has been significantly improved. To clearly state our research, new discussions have been added in the revision to make our findings more solid. The corresponding revisions concerning comments have been provided and highlighted in red in the revised manuscript and Supplementary Information. The corresponding responses of the reviewers are marked by blue words. The detailed revisions in the revised manuscript are listed on a separate page at the end of the response letter.

Responses to Reviewer # 1:

General Comments:

“The manuscript describes a novel Van der Waals all-day motion detector, which fully demonstrates the unique properties of Van der Waals materials. The work is very interesting and will further promote the application of Van der Waals materials for motion detection with simple structures and high performance. The results are convincing in the sense that they provide sufficient evidence that rippled-assisted phototransistors are capable of non-volatile memory, broad-spectrum image capture and image processing functions. The rippled interface of silicon nitride and MoS₂ monolayer used in the research have been characterized through various characterization methods, which makes the whole results much more solid. Meanwhile, broadband optoelectronic devices possess good performance with a simple structure, which derives great potential for integrated applications. Among the previously proposed works, the effort has been made to fabricate the light sensor with tunable responsivity using Van der Waals layered materials, which aims to achieve in-sensor-computing functionality. However, the complex structure with multiple layers of different materials limits the yield and thus the application. So far, the method to

manufacture optical sensors with sensing, memory and processing functions on a large scale has not been reported. Pang et al in this paper, for the first time, realized all-in-one optoelectronic devices on a large scale through a simple structure consisting of only a monolayer MoS₂ and a layer of silicon-rich silicon nitride. The authors provided sufficient and supportive data to demonstrate the all-day motion detector utilizing the broadband and the rippled-assisted optoelectronic devices, of which the simple structure shows great potential to overcome the limitations of yield and integration. The work will further promote the development of electronics and motion detectors by utilizing the extraordinary properties of Van der Waals semiconductors and the rippled-assisted interface. Hence, I recommend the paper for publication in Nature Communications if the authors will address the following questions:”

Response:

Thank the reviewer for carefully reading the manuscript and positive comments on our work ‘*this work is very interesting*’. Meanwhile, the reviewer has concerns about the memory property, response time, and responsivity. Based on these critical and constructive comments, the new experimental data have been added to the revised manuscript to enhance the readability. With the help of the reviewer, the whole manuscript has been largely improved. In the following, we will address all comments point-by-point and revised the manuscript. We hope that the revised manuscript would remove the reviewers’ concerns.

Comment 1:

“Response time is a key parameter indicating whether the device is suitable for continuously working in real circumstances. The author should provide the response time data with incident light with different wavelengths.”

Response:

Thank the reviewer for the valuable comment. We appreciate your careful consideration and are grateful for the opportunity to address your concerns. In response to your comments, we would like to provide additional clarification regarding the response time of the photodetectors. The response time of photodetectors is mainly composed of three parts¹: the photo-induced carrier generation time (on the order of

picosecond), transport time, and the external circuit time constant (on the order of nanosecond). Our analysis reveals that, among these factors, the transport time plays a pivotal role in determining the overall response time. Specifically, the transport time can be given as $\tau_{\text{tran}} = L^2/\mu V_{\text{bias}}$, where V_{bias} is the applied bias voltage, μ is the carrier mobility, and L is the length of the channel. Hence, the increment of carrier mobility can lead to a great reduction in transport time, which leads to a short response time suitable for continuously working in real circumstances.

In the previous manuscript, we presented the rippled-assisted optoelectronic (RAO) device featuring a monolayer of MoS₂, as illustrated in **Fig. 2a**. The response time of the RAO device is predominantly influenced by the mobility and channel length of the device. By incorporating a rippled interface of silicon-rich silicon nitride, we have successfully achieved a substantial reduction in response time. This improvement is demonstrated in **Fig. 1h**, where the photocurrent, generated under light stimuli lasting 10 ms, is showcased. Furthermore, we conducted extensive measurements of the photocurrent under varying light stimulus durations, ranging from 1 to 50 ms, as depicted in **Fig. R1.1a-f**. These results unequivocally validate the robustness of our device.

Fig. R1.1 | Current versus time curve with different durations of stimuli from 1 to 50 ms. The durations are selected as (a) 1 ms, (b) 2.5 ms, (c) 5 ms, (d) 20 ms, (e) 30 ms, (f) 50 ms, respectively.

Thank the reviewer for the constructive comments again, which helped us further test the extra characteristics of the devices. Moreover, your insightful suggestions not only guided us in testing the additional characteristics of our devices but also deepened our understanding of the optoelectronic response process

Revision:

We have reorganized Supplementary Figure. 10 to display the response time in the range from 1 to 50 ms (Page 35). Additionally, corresponding discussions have been incorporated into the revised manuscript as follows: *“Additionally, we conducted thorough extensive testing on the response time of the RAO device with light stimuli lasting from 1 to 50 ms (see Supplementary Information Fig. 10a-f). Remarkably, the RAO device exhibits proper functionality even under light pulses as short as 1 ms. Furthermore, the device continues to perform effectively when exposed to light pulses lasting up to 50 ms”*. (Page 7)

Reference:

1. George, G. & Krusius, J. P. Dynamic response of high-speed PIN and Schottky-barrier photodiode layers to nonuniform optical illumination. *J. Lightwave Technol.* **12**, 1387-1393 (1994).

Comment 2:

“ The authors should provide the responsivity data and its incident power dependence in visible and in infrared.”

Response:

Thank the reviewer for the valuable comment. In response, we have provided the responsivity data and its incident power dependence in both visible and infrared regions. **Fig. R1.2a** illustrates the responsivity of the first state under varying incident power at the wavelength of 405 nm, while **Fig. R1.2b** depicts the responsivity under varying incident power at the wavelength of 940 nm. Notably, as the power increases, the responsivity for both the visible and infrared lights decreases. Additionally, **Fig. R1.2c** showcases the responsivity with gate voltage ranging from 0 V to 15 V at 405 nm. The responsivity demonstrates a consistent trend with changing the incident power under different gate voltages, which can be as high as 181.66 A/W at the lowest power of 0.05 μ W.

Fig. R1.2 | The responsivity depends on multiple parameters. The responsivity of the first state under varying incident power at the wavelength of (a) 405 nm and (b) 940 nm, respectively. c, The responsivity of the device with different gate voltages at the wavelength of 405 nm.

Revision:

We have added **Supplementary Figure. 11** to display the responsivity characteristics of the RAO device. (Page 36)

Comment 3:

“For visible detectors, the data about electrical and optoelectronic properties of MoS₂, such as output curves are deficient. Please provide detailed data.”

Response:

We are grateful for the reviewer's valuable comments. We have incorporated the output curves of the MoS₂ device in multiple states into the revised manuscript, as depicted in Fig. R3. Notably, these experiments were conducted at $V_{BG} = 0V$, and the device states were modulated by 10 ms light stimuli at the wavelength of 520 nm. As illustrated in **Fig. R1.1-3**, the output curves distinctly exhibit a discernible gap between each pair. This observation underscores the effective tuning of channel conductance through light stimuli. The presented data further supports the robustness and versatility of our device under varying states induced by light stimuli.

Fig. R1.3 | Serial output curves of the device in various states. They were generated through light stimuli tuning at $V_{BG} = 0 V$. The device states were modulated by 10 ms light stimuli at the wavelength of 520 nm.

Revision:

We have added **Supplementary Figure. 14** to display the serial output curves tuned by light stimuli. (Page 39)

Comment 4:

“The detectors exhibit a quite low dark current. When concerning the area of the devices, the author should change to the current density. It will be more reasonable.”

Response:

We appreciate your valuable comment regarding the dark current of the detectors. In response to your suggestion, we have made the necessary correction with the consideration of the area of the devices, and changed the dark current to current density. The updated figure reflecting this change is now presented in **Fig. R1.4**.

Fig. R1.4 | The minimum and maximum drain currents (I_{DS}) per pixel width. Pixels are determined based on the respective positions.

Revision:

We have reorganized **Figure. 3 e-f** to display both the corrected maximum and minimum current of the RAO array (Page 16). Additionally, corresponding discussions have been incorporated into the revised manuscript as follows: “*The minimum source-to-drain currents per width of most phototransistors are below $10^{-5} \mu A/cm$ at the $V_{DS} = 0.5 V$, which approaches the limit of the test instrument. On the other hand, the maximum source-to-drain currents typically exceed $10^2 \mu A/cm$.*”. (Page 9)

Comment 5:

“As for the memory capability, the data illustrating the maintenance of the memory property is deficient. Please provide detailed data.”

Response:

Thank the reviewer for the valuable comment. In response, we have included additional data in the revised manuscript to address the raised concern. Specifically, in Fig. 2g-h of the manuscript, we incorporated the results of a stability test demonstrating the device's ability to operate continuously with robust stability during repetitions. Additionally, for the assessment of memory stability, we conducted long-term measurements of the channel current. The data, presented in **Fig. R1.5**, indicates that the storage state of the RAO device remains well-maintained for 10^4 s. This observation underscores the significant storage stability of both non-volatile PPC and non-volatile NPC.

Fig. R1.5 | The light induced memory stability of the RAO device. a, The non-volatile PPC after light stimuli. b, The non-volatile NPC after light stimuli. Both figures demonstrate remarkable memory stability.

Revision:

We have added **Supplementary Figure. 13** to display the memory stability of the RAO device (Page 39). Corresponding discussions have been incorporated into the revised manuscript as follows: “*We conducted long-term measurements of the channel current to assess this memory stability, which shows a remarkable retention ability (See Supplementary Fig.13)*”. (Page 5)

Comment 6:

“ What’s the difference between rippled interface and a flat interface? What enhancement of mobility does the rippled interface bring quantitatively?”

Response:

Thank the reviewer for the valuable comment. The mobility enhancement is attributed to the rippled interface of SiN_x^1 . We have included the relevant data in the revised manuscript, specifically in Fig. R1.6 a and d, which illustrates roughness and transfer curves for a flat SiN_x interface. By calculation, we obtain its mobility as $6.295 \text{ cm}^2 \text{ V}^{-1} \text{ s}^{-1}$. The introduction of the rippled interface as the dielectric layer results in a relative enhancement in mobility compared to devices on a flat SiN_x surface (Fig. R1.6c and Fig. R1.6f). It is worth noting that the mobility enhancement may decrease if the roughness is further reduced (Fig. R1.6b and Fig. R1.6e).

Fig. R1.6 | Roughness and transfer curves of devices on both flat and rippled silicon nitride interface. a, Roughness of the device on the flat interface by AFM. b, Roughness of device on the rippled interface with less roughness by AFM. c, Roughness of device on the pristine rippled interface by AFM. d, The transfer curve and mobility of the device on the flat interface. The mobility is $6.295 \text{ cm}^2 \text{ V}^{-1} \text{ s}^{-1}$. e, The transfer curve and mobility of the device on the rippled interface with less roughness. The mobility is $26.73 \text{ cm}^2 \text{ V}^{-1} \text{ s}^{-1}$. f, The transfer curve and mobility of the device on the pristine rippled interface. The mobility is $97.77 \text{ cm}^2 \text{ V}^{-1} \text{ s}^{-1}$, which shows a remarkable enhancement brought by the rippled interface.

Revision: We have added **Supplementary Figure. 4** to display the comparison of devices over flat and rippled silicon nitride interfaces. (Page 30)

Reference:

1. Ng, H. K. et al. Improving carrier mobility in two-dimensional semiconductors with rippled materials. *Nat. Electron.* **5**, 489-496 (2022).

Comment 7:

“In Fig. 2b, is the transfer curve clockwise or anti-clockwise? The direction of the transfer curve to some extent demonstrates the memory mechanism of this device. The author should mark the direction of the transfer curve.”

Response:

Thank the reviewer for the valuable comment. The transfer curve is anti-clockwise, and the mark of the direction has been added in the manuscript. The transfer curves all exhibit a memory window. As discussed in Section 2, Supplementary Information, the silicon-rich SiN_x dielectric layer contains numerous natural defects that are capable of trapping holes. The heavily p-doped silicon functions as a natural hole reservoir and can provide a sufficient number of holes for injection. While measuring the transfer curve of the RAO device, applying a positive gate voltage drives hole injection from the heavily p-doped silicon to the sr-SiN_x dielectric, which induces the electrons in the channel. This makes the channel conductance increase. The holes are kept before the voltage decreases, which makes the current larger than the current while the gate voltage increases. Also, applying a negative gate voltage releases the holes in the sr-SiN_x dielectric layer, which decreases the channel conductance and the channel current. In that case, the direction of the transfer curve is anti-clockwise.

Fig. R1.7 | Transfer curves of 262 pixels with a high yield of 80% at the $V_{DS} = 0.5$ V. The arrow shows the transfer curve as gate voltage increases. The deep blue curve represents a typical transfer curve for RAO processors, which shows a large memory window.

Revision:

We have reorganized **Figure. 3a** to display the anti-clockwise direction of the transfer curve. (Page 16)

Comment 8:

The authors also mentioned the dynamic range of their devices. Can they compare with existing works (e.g., Nature Electronics, 2022, 5, 84-91)?

Response:

Thank the reviewer for the valuable comment. The comparison of optical dynamic range under different wavelengths for intelligent image perception technology among previous works is shown in **Fig. R1.8**¹⁻¹¹. The reference mentioned in the comment (Nature Electronics, 2022, 5, 84-91) is represented by ref.7 in the figure below.

Fig. R1.8 | Comparison of optical dynamic range under different wavelengths for intelligent image perception technology in comparison to the previous works. The solid boxes represent intelligent image perception techniques that can be used for moving target detection. The RAO array achieves the MDR functions with the largest dynamic range and wavelength band among the previous works.

Revision:

We have reorganized Figure. 3i to display the comparison among the intelligent image processors and add references about this comparison. (Page 16)

Reference:

- Zhou, F. et al. Optoelectronic resistive random access memory for neuromorphic vision sensors. *Nat. Nanotechnol.* **14**, 776-782 (2019).
- Zhang, Z., Wang, S., Liu, C., Xie, R., Hu, W. & Zhou, P. All-in-one two-dimensional retinomorphic hardware device for motion detection and recognition. *Nat. Nanotechnol.* **17**, 27-32 (2022).
- Wang, H. et al. A Retina-Like Dual Band Organic Photosensor Array for Filter-Free Near-Infrared-to-Memory Operations. *Adv. Mater.* **29**, 1701772 (2017).
- Wang, C.-Y. et al. Gate-tunable van der Waals heterostructure for reconfigurable neural network vision sensor. *Sci. Adv.* **6**, eaba6173
- Seo, S. et al. Artificial optic-neural synapse for colored and color-mixed pattern recognition. *Nat. Commun.* **9**, 5106 (2018).
- Mennel, L., Symonowicz, J., Wachter, S., Polyushkin, D. K., Molina-Mendoza, A. J. & Mueller, T. Ultrafast machine vision with 2D material neural network image sensors. *Nature* **579**, 62-66 (2020).
- Liao, F. et al. Bioinspired in-sensor visual adaptation for accurate perception.

- Nat. Electron.* **5**, 84-91 (2022).
8. Lai, H. et al. Photoinduced Multi-Bit Nonvolatile Memory Based on a van der Waals Heterostructure with a 2D-Perovskite Floating Gate. *Adv. Mater.* **34**, 2110278 (2022).
 9. Jang, H. et al. An Atomically Thin Optoelectronic Machine Vision Processor. *Adv. Mater.* **32**, 2002431 (2020).
 10. Jang, H. et al. In-sensor optoelectronic computing using electrostatically doped silicon. *Nat. Electron.* **5**, 519-525 (2022).
 11. Chen, J. et al. Optoelectronic graded neurons for bioinspired in-sensor motion perception. *Nat. Nanotechnol.* **18**, 882-888 (2023).

Responses to Reviewer # 2:

In this study, the authors show a $2 \times 2 \text{ mm}^2$ integration of what they claim a rippled-assisted optoelectronic (RAO) array (18×18 pixels) to mimic the characteristics of snake vision systems for all-day motion detection and recognition. The claim of a snake vision almost reads like a buzzword claim. Detection of light of 940 nm is nothing new. In fact MoS₂ has been shown to possess both positive and negative PPC in even broader wavelength ranges than what is reported here using other substrates such as p doped Ge. There are several claims in the paper that are not evidenced properly. Nor was I able to really identify what the key advance here is.

There are multiple issues that need to be addressed:

Response:

We appreciate your insightful comments on our research. Among the previous papers related to broadband detection based on MoS₂ and Ge, the majority have demonstrated capabilities in detecting a range from 406nm to 1550nm^{1,2}. For instance, the work in the Science Advances paper¹ demonstrated broadband detection utilizing a MoS₂ and Ge heterostructure, exhibiting a positive photo response without memory functionality. Similarly, the Advanced Functional Materials paper² showcased remarkable broadband detection based on MoS₂ and Ge heterostructure with positive and negative photo response tuned by incident wavelength. However, the MoS₂/Ge heterostructure based photodetectors lack the capability of storing the optical signal in a non-volatile manner, which is critical for applications such as motion detection and in-sensor processing.

In our work, macroscale ($2 \times 2 \text{ mm}^2$) integration of a rippled-assisted optoelectronic (RAO) array (18×18 pixels), possessing negative and positive optical detection as well as memory and computation capabilities, mimics the characteristics of snake vision systems for all-day motion detection and recognition. Specifically, the significance of this work includes the following:

(1). **Rippled-assist in increasing mobility and dynamic storage range.** The RAO processor not only enables continuously reconfigurable non-volatile positive and negative photoconductance (NV-PPC and NV-NPC), but also enhances room-temperature mobility and optical storage dynamic range. Specifically, the RAO array achieves an extensive optical storage dynamic range exceeding 10^6 , and exceptionally

high room-temperature mobility up to $406.7 \text{ cm}^2 \text{ V}^{-1} \text{ s}^{-1}$, four times higher than the International Roadmap for Device and Systems (IRDS) 2028 target.

(2). **Broadband spectrum for all-day MDR functions.** Inspired by the snake vision systems design, a novel approach has been developed for motion imaging and recognition in the dark environment that extends beyond only visible radiation. Each RAO processor combines the functions of the snake's eye and pit organs, enabling the array to detect moving targets in both bright and dark environments. Importantly, the spectral range of the RAO array covers visible to near-infrared (405 nm to 940 nm), achieving detection and recognition of a man riding an electrical bicycle in both bright and dark environments. To our knowledge, this is the first demonstration of all-day motion detection and recognition functions.

(3). **A simple technique to achieve a high level of integration for all-day MDR functions.** Floating gate devices based on multi-layer stacked heterostructures make it difficult to achieve large-scale integration of positive and negative optical storage devices due to the manufacturing process complex. In our work, a parallel atomically thin MoS_2 optoelectronic array (18×18 pixels) was fabricated on a specially treated commercial sr-SiN_x substrate to achieve non-volatile positive and negative photoconductance. Moreover, the rippled MoTe_2 processor also demonstrates similar continuously reconfigurable non-volatile positive and negative photoconductance, illustrating the consistent optoelectronic properties of the rippled structure.

The distinction between reported vision sensors²⁻⁴ and our RAO devices has been shown in **Fig. R2.1**. Notably, the RAO device demonstrates the capability to process both static and moving objects, effectively achieving all-in-one functionality. Unlike vision sensors that necessitate a complex assembly of functional modules to accomplish motion target processing⁵, our RAO device represents a significant advancement in the detection and recognition of moving objects.

We have revised the manuscript according to your suggestions and believe that these revisions have improved the paper. With the help of the reviewer, the whole manuscript has been largely improved. In the following, we will address all comments point-by-point and revised the manuscript. We hope that the revised manuscript would remove the reviewers' concerns.

	Property	Ability	Applications
The reported vision sensors: without time differential ability (Sci. Adv. 2021, 7, eabj2521; Adv. Funct. Mater. 2022, 32, 2110181; Nature, 2020, 579 (7797) : 62-66.)		[ ] Positive and negative photoresponse tuned by incident wavelength	[ ] Static image processing: inverse; edge enhancement; contrast correction; [ ] Image recognition with extra memory;
Our RAO devices : with time differential ability		[ ] Positive and negative photoresponse regardless incident wavelength [ ] Non-volatile photoresponse [ ] Photo-modulated linear multi-states	[ ] Static image processing: inverse; edge enhancement; contrast correction; [ ] Image recognition with memory; [ ] Moving objects detection and recognition;

Fig. R2.1 | Comparison of the reported vision sensors and our RAO devices.

Revision:

We've revised the abstract (Page 2) and added several citations about previous works about broadband detection based on MoS₂ in the introduction. (Page 3)

Reference:

1. Hwang, A. et al. Visible and infrared dual-band imaging via Ge/MoS₂ van der Waals heterostructure. *Sci. Adv.* **7**, eabj2521
2. Wang, B. et al. Mixed-Dimensional MoS₂/Ge Heterostructure Junction Field-Effect Transistors for Logic Operation and Photodetection. *Adv. Funct. Mater.* **32**, 2110181 (2022).
3. Trujillo Herrera, C. & Labram, J. G. A perovskite retinomorph sensor. *Appl. Phys. Lett.* **117**, 233501 (2020).
4. Zhou, F. et al. Optoelectronic resistive random access memory for neuromorphic vision sensors. *Nat. Nanotechnol.* **14**, 776-782 (2019).
5. Posch, C., Serrano-Gotarredona, T., Linares-Barranco, B. & Delbruck, T. Retinomorph Event-Based Vision Sensors: Bioinspired Cameras With Spiking Output. *Proc. IEEE* **102**, 1470-1484 (2014).

Comment 1:

Why were measurements done on mechanically exfoliated MoS₂ when the authors have access to 18 by 18 pixels of CVD grown films?

Response:

Thank the reviewer for the valuable comment. We performed a comprehensive assessment of MoS₂ devices obtained through both chemical vapor deposition (CVD) synthesis and mechanical exfoliation (ME), by using the same rippled substrate. The comparison between devices fabricated from CVD and ME films, featuring identical channel length and width, holds significant value in elucidating the consistency and performance of these materials in electronic applications. Here are some key points highlighted:

(1). **Consistency in electrical properties:** The measurements on both ME and CVD devices exhibit a high level of consistency in electrical properties, as depicted in Fig. R2.2. The measured mobilities of 97.77 cm²V⁻¹s⁻¹ for CVD MoS₂ and 89.84 cm²V⁻¹s⁻¹ for ME MoS₂ provide quantitative insights into the charge carrier mobility of these materials. The small variation in mobility suggests that both synthesis methods yield devices with similar charge transport characteristics. The overlapping memory window and a relatively small variation in mobility (9%) between CVD and ME devices underscore the comparable electrical performance of the two synthesis methods.

Fig. R2.2 | Transfer curves and mobility of monolayer MoS₂ transistors by both CVD and ME. The transistors show a great consistency in electrical properties such as memory window and mobility. The mobility for the device of CVD MoS₂ is 97.77 cm²V⁻¹s⁻¹ and 89.84 cm²V⁻¹s⁻¹ for the one by mechanically exfoliated film, of which the variation is less than 9%.

(2). **Consistency in optoelectronic characteristics:** The observation of a common trend in both CVD and ME devices, where responsivity decreases as effective incident light power increases (Fig. R2.3a-b), indicates a consistent behavior in the

optoelectronic response of MoS₂ devices. Additionally, concerning non-volatile photocurrent, both devices demonstrate similarity under identical test conditions for both positive and negative photocurrent, as illustrated in **Figure R2.4a-b**. This uniformity in optoelectronic responses between CVD and ME devices further emphasizes the comparable performance of these devices in terms of their behavior under varying light power conditions and continuous light stimuli.

Fig. R2.3 | Responsivity of both CVD and ME devices. a, The responsivity of the CVD device with different gate voltages at a wavelength of 405 nm. As the light power increases, the responsivity decreases. b, The responsivity of the ME device with different gate voltages at a wavelength of 405 nm. It shows the same trend as the CVD device when incident power increases.

Fig. R2.4 | Non-volatile positive and negative photocurrent of CVD and ME devices. a, Non-volatile positive photocurrent of CVD and ME devices with continuous light stimuli. b, Non-volatile negative photocurrent of CVD and ME devices with

continuous light stimuli, where the incident power is fixed at 13.5 μW and the time duration of stimuli is 20 ms at the wavelength of 520 nm.

In summary, we express our gratitude to the reviewers for their constructive comments. Their insights have been instrumental in validating the consistency of the electrical properties of MoS_2 obtained through different synthesis methods. Additionally, their feedback has provided valuable insights into the optoelectronic properties of our study. Thank you for your thoughtful and constructive feedback.

Revision:

We have added Section 9, Supplementary and **Supplementary Fig. 22-24** (Page 47-49) to illustrate the consistency between ME and CVD devices. Corresponding discussions have been incorporated into the revised manuscript as follows: “*We performed a comprehensive assessment of MoS_2 characteristics obtained through both chemical vapor deposition (CVD) synthesis and mechanical exfoliation (ME). The comparison between devices fabricated from CVD and ME films, featuring identical channel length and width, holds significant value in elucidating the consistency and performance of these materials in electronic applications (shown in **Supplementary Fig. 22-24**)*”. (Page 8)

Comment 2:

There is no evidence shown to support the claim of 18 by 18 pixels and such large area growth.

Response:

Thank you for your valuable suggestions. We demonstrate the accuracy of 18 by 18 pixels through experiments concerning the large-area device growth, array fabricates processes, microscope and SEM images, and the electrical and optoelectronic performance of the MDR array.

(1). **The synthesis process of the CVD monolayer MoS_2 :** The monolayer MoS_2 sample supplied by Six Carbon Shenzhen was synthesized by CVD technique on the sapphire substrate and then transferred onto the SiO_2 substrate. The specific process flow for large area growth of MoS_2 was obtained from Six Carbon Shenzhen. MoO_3 (purity over 99.999%) is used as the Mo source and solid sulfur is used as S source. The

process is taken in the Ar atmosphere in the tube furnace of which the diameter is 80 mm with a double temperature zone. MoO₃ is heated to 650 °C and solid sulfur is heated to 180 °C with pressure 4000 Pa in the tube. The process takes 10 minutes and the MoS₂ is synthesized on the sapphire surface. Then the monolayer MoS₂ is carefully transferred into the target chip.

(2). **The fabrication processes are as follows:** For the RAO array, the monolayer MoS₂ sample is transferred onto the SiO₂ substrate. Firstly, the MoS₂ on the SiO₂ substrate was spin-coated with polystyrene (PS) and kept at 90 °C for 10 minutes. Then both the monolayer film and the PS were separated from the SiO₂ substrate using deionized water, and transferred onto a polydimethylsiloxane (PDMS). After that, the PDMS substrate carrying the MoS₂ film was transferred onto the specially treated sr-SiN_x substrate by the fixed-point transfer technology keeping 100 °C during the transfer process. The substrate, monolayer MoS₂ and the PS layer were then soaked into the methylbenzene to remove the polystyrene (PS) layer. Source electrodes were patterned using laser direct writing (LDW) technology (Micro-Writer ML3) and subsequently deposited by EBE. The separation layer (SiO₂) at the junction of the source and drain electrodes was patterned using LDW technology and deposited by physical vapor deposition (PVD, Lesker PVD 75). Then, drain electrodes were patterned using LDW technology and deposited by EBE. The MoS₂-channel region was obtained by O₂ plasma etching. After the lift-off process, the RAO array was successfully fabricated.

(3). **Microscope and SEM images of the MDR array.** We've captured images of the MDR array comprising 18 by 18 pixels with both a microscope and a scanning electron microscope (SEM). The microscope view of the RAO array is shown in **Fig. R2.5a-c**, where the gold part represents the source and drain contacts. The dashed box in the **Fig. R2.5b** shows the channel region of the MoS₂ monolayer. The SEM view of the RAO array is depicted in **Fig. R2.5 d-f**. As shown in **Fig. R2.5f**, the orange dashed box represents one of the SiO₂ regions, the blue dashed box represents the channel region, and the green dashed box represents part of the metal contact.

Fig. R2.5 | Images of the MDR array at different magnifications. a, Enlarged microscope view of the 18×18 RAO array by 200x. b, Enlarged microscope view of the 6 RAO devices by 500x. c, More enlarged microscope view of single RAO device by 1000x. d, SEM view of the 18×18 RAO array. e, Enlarged SEM view of the 18×18 RAO array magnified by 221x. f, Enlarged SEM view of the 18×18 RAO array magnified by 626x.

(4). **Electrical and optoelectrical characteristics of the MDR array.** Both electrical and optoelectrical properties of 18×18 RAO array are measured, and some of the statistical data are demonstrated in **Fig. R2.6a-c**. The uniformity of the maximum and minimum current of the transfer curves are shown in **Fig. R2.6a-b**, where the majority of the currents fall within an order of magnitude for each case. Most phototransistors exhibit minimum source-to-drain currents below 10^{-13} A at $V_{DS} = 0.5$ V, approaching the limit of the test instrument. Conversely, the maximum source-to-drain currents typically exceed 10^{-5} A. In regard to the optoelectronic properties of the MDR array, Fig. R2.6c illustrates the data of both the NV-PPC and NV-NPC for the RAO array under periodic and continuous optical pulses. The information is visualized in a box chart, illustrating the distribution, mean, and median for each state. Despite the large amounts of pixels, the relatively low overlap between each two adjacent states indicates excellent stability and functionality, rendering the RAO array suitable for MDR applications.

Fig. R2.6 | Array statistics of both electrical and optoelectrical characteristics. a, Minimum I_{DS} of each pixel according to its position. **b,** Maximum I_{DS} of each pixel according to its position. **c,** State statistics induced by optical stimuli for both NV-PPC and NV-NPC. The overlapping region is hardly observed between the two adjacent states, indicating outstanding uniformity.

Revision:

We've added the synthesis method of CVD monolayer MoS_2 and a more detailed fabrication process in the Method. Corresponding method have been incorporated into the revised manuscript as follows: *“the monolayer MoS_2 sample supplied by Six Carbon Shenzhen was synthesized by CVD technique on the sapphire substrate and then transferred onto the SiO_2 substrate. The specific process flow for large area growth MoS_2 was obtained from Six Carbon Shenzhen. MoO_3 (purity over 99.999%) is used as the Mo source and solid sulfur is used as S source. The process is taken in the Ar atmosphere in the tube furnace of which the diameter is 80 mm with a double temperature zone. MoO_3 is heated to 650 °C and solid sulfur is heated to 180 °C with pressure 4000 Pa in the tube. The process takes 10 minutes and the MoS_2 is synthesized on the sapphire surface. Then the monolayer MoS_2 will be carefully transferred into target chip like Si/ SiO_2 substrate”* (Page 22).

Comment 3:

In the supporting information, suddenly MoTe_2 appears out of nowhere. Is there a rationale for that.

Response:

Thank the reviewer for the valuable comment. Besides monolayer MoS_2 , we've tested the property of MoTe_2 to better evaluate the enhancement induced by the rippled

interface. This result shows that the rippled sr-SiN_x dielectric layer has good generality and flexibility for providing the functionality of both positive and negative photo response and memory effect for MoTe₂ materials.

(1). **Memory capability.** The observation of a memory gap in the transfer curve represents the memory capacity brought by the sr-SiN_x dielectric. Importantly, both MoS₂ and MoTe₂ devices on this specifically treated dielectric exhibit a remarkable memory gap (**Fig. R2.7a-b**). It is also noteworthy that, owing to the ambipolar feature of MoTe₂, the transfer curve of the MoTe₂ device displays an additional, smaller memory gap. This underscores the versatility of the sr-SiN_x layer in imparting memory functionality.

Fig. R2.7 | Transfer curves of MoTe₂ and MoS₂ device. a, Transfer curve of MoTe₂ device at the $V_{DS} = 0.1$ V, where a huge memory gap and a smaller memory gap are observed. b, Transfer curve of MoS₂ device at the $V_{DS} = 0.1$ V, where a huge memory gap is observed.

(2). **Positive and negative photo response.** As shown in Fig. R2.8a-b, the MoS₂ device shows the features of both positive photon response (PPC) and negative photo response (NPC) in the microamp scale. Similarly, the MoTe₂ device shows the same features of PPC and NPC in the scale of nanoamp, which is illustrated in Fig. R2.8c-d. This indicates the generality of this sr-SiN_x layer for providing the functionality of positive and negative photo response.

Fig. R2.8 | Positive and negative photo response for both MoS₂ and MoTe₂ devices.

a, Cumulative non-volatile positive photocurrent of MoS₂ device with progressive multilevel states under periodic optical pulses. b, Cumulative non-volatile negative photocurrent of MoS₂ device with progressive multilevel states under periodic optical pulses. c, Cumulative non-volatile negative photocurrent of MoTe₂ device with progressive multilevel states under periodic optical pulses. d, Cumulative non-volatile positive photocurrent of MoTe₂ device with progressive multilevel states under periodic optical pulses. For a-d, the lasting time is 5 ms and the period is 1 s at the wavelength of 520 nm. All devices are fabricated by using mechanically exfoliated films.

Revision:

We've revised Section 10, Generality and flexibility of sr-SiN_x, Supplementary and added **Supplementary Fig. 25-26** (Page 50-51) to display the generality and flexibility of the sr-SiN_x platform for providing functions of both PPC, NPC and memory. Corresponding discussions have been incorporated into the revised manuscript as follows: *“To further verify the functionality of the specially treated sr-SiN_x, we've tested the property of MoTe₂ to better evaluate the enhancement induced by the rippled interface. The rippled MoTe₂ device demonstrates the same continuously reconfigurable NV-PPC and NV-NPC, illustrating the rippled sr-SiN_x dielectric layer has good generality and flexibility for providing the functionality of both positive and*

negative photo response and memory capacity for some types of Transition Metal Dichalcogenides (See Supplementary Section 10 and Supplementary Fig.25-31)".
(Page 8)

Comment 4:

The authors claim to have achieved very high room temp mobilities. No information is provided how these were calculated and what mechanism allows such high mobilities.

Response:

We thank the reviewer's comment. As mentioned in the manuscript, we performed a statistical analysis of the carrier mobility calculated by using the formula: $\mu = \frac{dI_{DS}}{dV_{GS}} \times [L/W C_i V_{DS}]$, where L , W and C_i are the channel length, the channel width, and the capacitance between the channel and the back gate per unit area, respectively. In the following context, we'll provide information about this high mobility from two aspects theory analysis and experiment verification.

1. Theory analysis.

In the Nature Electronics paper¹, it has demonstrated that lattice distortions have the potential to reduce electron-phonon scattering in 2D materials, consequently improving charge carrier mobility. The presence of ripples in the MoS₂ induced by a bulged substrate brings about a change in the dielectric constant and suppresses photon scattering, resulting in enhanced mobility.

Also, a first-principles calculation has been performed to illustrate the mechanism of the enhanced carrier mobility of monolayer MoS₂. The calculation details are as follows:

Our first-principles calculations were performed with Vienna Ab-initio Simulation Package (VASP)^{2, 3} using the projector augmented wave (PAW)⁴ method. The exchange-correlation interactions were handled with Perdew-Burke-Ernzerhof (PBE)⁵ functional. The cutoff energy for the plane-wave expansion was set to 400 eV in the whole process. The atom positions were relaxed until forces on them were less than 10^{-2} eV/Å. The rippled MoS₂ was constructed by sin function based on the $1 \times 3 \times 1$

supercell of orthogonal MoS₂. The rippled heights were selected 1, 2, and 3 Å, respectively, corresponding to the vertical distance between the highest and lowest Mo atoms. For the first Brillouin zone sampling, Γ -centered Monkhorst-Pack k-point meshes of $8 \times 2 \times 1$ and $16 \times 3 \times 1$ were used to perform structural relaxation and optical properties calculations. In addition, a vacuum layer larger than 15 Å was added to reduce the mirror interactions.

To elucidate the impact of substrate-induced bulges on the dielectric constant, we conducted dielectric function calculations for monolayer MoS₂ supercells employing four idealized models: flat MoS₂ and corrugated MoS₂ with curvature heights of 1 Å (Rippled-1 Å), 2 Å (Rippled-2 Å), and 3 Å (Rippled-3 Å). Progressing from flat MoS₂ to corrugated-1 Å, and from Rippled-1 Å to Rippled-3 Å, the relationship reveals an increasing curvature height, resulting in non-uniform strain within the twisted MoS₂. The curvature height is defined as the difference in height between the highest and lowest Mo atoms, as illustrated in **Fig. R2.9a-d**. Energy band structure calculations for these four model structures are presented in **Fig. R2.9e-h**, indicating a progressively diminishing energy band in MoS₂ as curvature increases.

To further investigate the relationship of the mobility of MoS₂, the mobility μ can be obtained by the following⁶:

$$\mu = \frac{M_M M_X A t^2 \varepsilon^2 (\hbar \omega)^2}{16 \pi^2 e^3 n m^* Z_{MB}^2 (\sqrt{M_M} + \sqrt{M_X})^2} \quad (1)$$

where the M_M , and M_X are the atomic mass of metal and chalcogen atoms, A is the area of the unit cell, n is the Bose–Einstein distribution, t is the effective thickness, ε is the in-plane optical dielectric constant, Z_{MB} is the Born effective charge of the M. t and ε can be approximated by the bulk dielectric constant and the interlayer distance in the bulk material. In accordance with Equation (1), it is observed that the material's mobility is directly proportional to ε^2 . The calculations of the real part of the dielectric function (ε_1) for the aforementioned four model structures are presented in **Fig. R2.10** and **Fig. R2.11**, with the static permittivity of MoS₂ determined by the value at $E \rightarrow 0$. It is noteworthy that these calculations exclude the ionic contribution, as its impact is negligible in comparison to the electronic contribution. The results demonstrate that

both the in-plane and out-of-plane dielectric constants of MoS₂ exhibit an increase with the height of curvature. Consequently, the ripples in the MoS₂ caused by the bulged substrate lead to a change in the dielectric constant, which leads to mobility enhancement.

Fig. R2.9 | 3D views and electronic band structures of rippled MoS₂ crystal from different curvature height. a-d, 3D views of (a) flat-MoS₂ and rippled-MoS₂ with (b) 1 Å, (c) 2 Å and (d) 3 Å curvature heights. e-h, Electronic band structures of (e) flat-MoS₂ and rippled-MoS₂ with (f) 1 Å, (g) 2 Å and (h) 3 Å curvature heights.

Fig. R2.10 | First-principles calculations of the dielectric function's real part (ϵ_1)

for monolayer MoS₂. a-d, Flat-MoS₂ and rippled-MoS₂ with curvature heights of 1 Å, 2 Å, and 3 Å.

Fig. R2.11 | Comparison of the static dielectric constants of monolayers of MoS₂ with varying curvature heights (flat-rippled, rippled-1 Å, rippled-2 Å, and rippled-3 Å), calculated based on first principles. $\epsilon_{||}$ and ϵ_{\perp} represent in-plane and out-of-plane dielectric constants of MoS₂, respectively.

2. Experiment verification.

To better evaluate the enhancement brought by the bulged interface, we've fabricated devices with flat, less rippled and rippled interfaces. By calculation, we get their mobility as $6.295 \text{ cm}^2 \text{ V}^{-1} \text{ s}^{-1}$, $26.73 \text{ cm}^2 \text{ V}^{-1} \text{ s}^{-1}$ and $97.77 \text{ cm}^2 \text{ V}^{-1} \text{ s}^{-1}$, respectively. The degradation is obvious as the roughness decreases (**Fig. R2.9a-c**). The introduction of the rippled interface as the dielectric layer results in a relative enhancement in mobility compared to devices on a flat SiN_x surface.

Further, to ensure the fidelity of the mobility calculations, we conducted a statistical analysis of the mobility of two different device arrays with rippled and less rippled interfaces. The mobilities distribution is shown in **Fig. R2.12a-b**. For the rippled interface, the maximum and average mobilities are 406.7 and $116.5 \text{ cm}^2 \text{ V}^{-1} \text{ s}^{-1}$, respectively. In contrast, devices over an interface with less roughness exhibit maximum and average mobilities of 78.5 and $31.3 \text{ cm}^2 \text{ V}^{-1} \text{ s}^{-1}$, respectively, which are significantly smaller than those observed for devices over rippled sr-SiN_x dielectric.

This strongly proves that our device has remarkable mobility enhancement characteristics.

Fig. R2.12 | Roughness and transfer curves of devices on both flat and rippled silicon nitride interface. a, Roughness of the device on the flat interface by AFM. b, Roughness of device on the rippled interface with less roughness by AFM. c, Roughness of device on the pristine rippled sr-SiN_x interface by AFM. d, The transfer curve and mobility of the device on the flat interface. The mobility is 6.295 cm² V⁻¹ s⁻¹. e, The transfer curve and mobility of the device on the rippled interface with less roughness. The mobility is 26.73 cm² V⁻¹ s⁻¹. f, The transfer curve and mobility of the device on the pristine rippled interface. The mobility is 97.77 cm² V⁻¹ s⁻¹, which shows a remarkable enhancement brought by the rippled interface.

Fig. R2.13 | Mobility distribution of device array with dielectric of different

interface roughness. a, Mobility distribution of device array with pristine sr-SiN_x and the Gauss fit of the distribution. The maximum and average mobilities are 406.7 and 116.5 cm² V⁻¹ s⁻¹. b, Mobility statistics of the array with less roughness dielectric and the Gauss fit of the distribution. The maximum and average mobilities are 78.5 and 31.3 cm² V⁻¹ s⁻¹.

Revision:

We have added Section 2, Mechanism of mobility enhancement and first-principles calculation, Supplementary and **Supplementary Fig. 7-9** to show the details of the first-principles calculation of mobility enhancement (Page 32-34). Also, we've added **Supplementary Figure. 4** to display the comparison of devices over flat and rippled silicon nitride interfaces (Page 30). Corresponding discussions have been incorporated into the revised manuscript as follows: "*Besides, a first-principles calculation is performed to obtain this mobility enhancement of monolayer MoS₂, of which the detailed discussion is shown in Supplementary Section 2*". (Page 6)

Reference:

1. Ng, H. K. et al. Improving carrier mobility in two-dimensional semiconductors with rippled materials. *Nat. Electron.* **5**, 489-496 (2022).
2. Kresse, G. & Furthmüller, J. Efficient iterative schemes for ab initio total-energy calculations using a plane-wave basis set. *Phys. Rev. B* **54**, 11169-11186 (1996).
3. Kresse, G. & Furthmüller, J. Efficiency of ab-initio total energy calculations for metals and semiconductors using a plane-wave basis set. *Comput. Mater. Sci.* **6**, 15-50 (1996).
4. Blöchl, P. E. Projector augmented-wave method. *Phys. Rev. B* **50**, 17953-17979 (1994).
5. Perdew, J. P., Burke, K. & Ernzerhof, M. Generalized Gradient Approximation Made Simple. *Phys. Rev. Lett.* **77**, 3865-3868 (1996).
6. Cheng, L. & Liu, Y. What Limits the Intrinsic Mobility of Electrons and Holes in Two Dimensional Metal Dichalcogenides? *J. Am. Chem. Soc.* **140**, 17895-17900 (2018).

Comment 5:

The explanation of positive and negative photocurrent is very shallow and highly speculative without any evidence or theoretical modeling. If the explanation provided is to be taken at face value, then every material with a direct bandgap and a Schottky

barrier with the substrate will show a similar behaviour which raises questions about the veracity of the explanation provided.

Response:

Thank you for your valuable comment. We have supplemented and improved the explanation of the mechanism. The silicon-rich SiN_x dielectric layer contains numerous natural defects that are capable of trapping holes¹⁻³. The Energy-dispersive X-ray spectroscopy test of the sr- SiN_x dielectric layer is performed to obtain the ratio of the Si and N, which is shown in **Fig. R2.14**. We've tested 10 locations on the sr- SiN_x substrate and the results are shown in the **Table. R1**. The average Si/N ratio is 1.397:1. It is worth mentioning that heavily p-doped silicon functions as a natural hole reservoir and can provide a sufficient number of holes for injection. During the writing process, a +20 V gate voltage is applied to drive hole injection from the heavily p-doped silicon to the sr- SiN_x dielectric. The injected holes are subsequently trapped by hole-trapping centers, which act as positive local gates that enhance the electron concentration and channel conductance, thereby corresponding to the memory of the low resistance state, which is shown in **Fig. R2.15a-b**. For the light erase process, a fixed -3 V gate voltage is applied accompanied by the optical stimulus, in which the holes in the dielectric are released. The reduction of stored holes leads to a smaller channel current, resulting in the negative photo conductance (NPC) in a non-volatile manner. The carrier distribution and band diagram of NPC is shown in **Fig. R2.15c**.

During the erasing process, a fixed -20 V gate voltage is applied to release the trapped holes, leading to a decrease in channel conductance that corresponds to the memory of the high resistance state (shown in **Fig. R2.15d-e**). As for the light write process, a fixed 3 V gate voltage is applied accompanied by the optical stimulus, in which the holes in the dielectric are trapped as the photon carriers generated by optical stimuli overcome the $\text{MoS}_2/\text{sr-SiN}_x$ interface barrier (shown in **Fig. R2.15f**), leading to the positive photo conductance (PPC) in a non-volatile manner.

Further, as illustrated in **Fig. R2.7** and **Fig. R2.8**, both MoTe_2 and MoS_2 have this property of positive and negative photo response, and memory. This indicates the sr- SiN_x can be used to provide this unique function of positive and negative photo response, and the memory capability, for TMD materials to some extent, which proves the explanation we provide.

Fig. R2.14 | Energy-dispersive X-ray spectroscopy of the sr-SiN_x layer. The first peak from the left represents the N element, and the second peak represents the Si element.

Fig. R2.15 | Positive and negative photoconductive mechanisms. **a-b**, The writing process causes the device to transition to a low-resistance state. **c**, Light erase process and negative photo conductance. **d-e**, The erasing process makes the device return to a high resistance state. **f**, Light write process and positive photo conductance.

Position index	Si percentage (%)	N percentage (%)
1	57.98	42.02
2	57.60	42.40
3	58.90	41.10
4	58.43	41.57
5	58.35	41.65
6	59.59	40.41
7	57.38	42.62
8	59.40	40.60
9	58.08	41.92
10	57.14	42.86
Average	58.285	41.715

Table. R1 | Si and N element ratio at ten different positions of the sr-SiN_x substrate in energy-dispersive X-ray spectroscopy.

Revision:

We have added **Supplementary Fig. 6** and **Supplementary Table. 1** to show the Si and N ratio in the sr-SiN_x dielectric layer (Page 31). Besides, We have revised Section 4, Memory and PPC, NPC mechanisms of the RAO processor, Supplementary and added **Supplementary Fig. 12** to better illustrate the mechanisms of PPC and NPC (Supplementary Page 37-38). Corresponding discussions have been incorporated into the revised manuscript as follows: *“For the light erase process, a fixed -3 V gate voltage is applied accompanied by the optical stimulus, in which the holes in the dielectric are released. The reduction of stored holes leads to a smaller channel current, resulting in negative photo conductance (NPC). The carrier distribution and band diagram of NV-PPC and NV-NPC are shown in **Supplementary Fig. 12**”*. (Page 5)

Reference:

1. Gritsenko, V. A., Perevalov, T. V., Orlov, O. M. & Krasnikov, G. Y. Nature of traps responsible for the memory effect in silicon nitride. *Appl. Phys. Lett.* **109**, 062904 (2016).
2. Pei, Z., Chung, A. & Hwang, H. L. Nonvolatile polycrystalline silicon thin film transistor memory using silicon-rich silicon nitride as charge storage layer. *Appl. Phys. Lett.* **90**, 223513 (2007).
3. Wrazien, S. J., Zhao, Y., Krayner, J. D. & White, M. H. Characterization of SONOS oxynitride nonvolatile semiconductor memory devices. *Solid-State Electron.* **47**, 885-891 (2003).

Overall, the study is very premature, shows some results but the claims do not really stack up in terms of existing literature in terms of advancement shown. Moreover, key claims are not backed by solid evidence and has a lot of hyperbole in the abstract and introduction. I would suggest that the authors look at other papers on MoS₂ that have shown better photodetection spectral range while showing positive and negative PPC. Unfortunately, I do not think that this paper nevertheless reports an advance that merits publication in this journal and cannot therefore recommend publication.

Response:

Thank you for your comments. We appreciate your constructive criticism and have carefully reevaluated our study in light of your suggestions. The manuscript has been revised to address the concerns raised, including a thorough examination of existing literature to ensure our claims align appropriately with the advancements shown by other studies on MoS₂. We have also taken steps to provide more solid evidence and have tempered any hyperbole in the abstract and introduction. We believe these revisions enhance the overall quality and credibility of our work. Your feedback has been invaluable, and we are grateful for the opportunity to improve our manuscript based on your recommendations.

Responses to Reviewer # 3:

The manuscript titled "Non-volatile Rippled-Assisted Optoelectronic Array for All-day Motion Detection and Recognition" explores a novel approach to motion detection and recognition (MDR) by integrating a rippled-assisted optoelectronic (RAO) array. The study addresses the challenges associated with conventional CMOS image sensors and presents a unique device capable of continuous reconfigurable non-volatile positive and negative photoconductance, enhanced mobility, and a wide optical storage dynamic range. The RAO array aims to achieve MDR in various lighting conditions, including dark environments. Overall it is a good paper and worthy of publication after several comments below are addressed.

Response:

Thank the reviewer for carefully reading the manuscript and positive comments on our work '*it is a good paper and worthy of publication*'. Meanwhile, the reviewer has concerns about mobility, contact resistance and image capture. Based on these critical and constructive comments, the new experimental data have been added to the revised manuscript to enhance the readability. With the help of the reviewer, the whole manuscript has been largely improved. In the following, we will address all comments point-by-point and revised the manuscript. We hope that the revised manuscript would remove the reviewers' concerns.

Comments 1:

I think the major result is high mobility, not necessarily all the motion detection and recognition. I would suggest the authors do a much more thorough assessment of the mobility and provide standard deviation.

Response:

Thank you for your valuable comment. In the following context, we'll make a thorough assessment of the mobility two aspects of theory analysis and experiment verification.

1. Theory analysis. In the Nature Electronics paper¹, it was demonstrated that lattice distortions have the potential to reduce electron-phonon scattering in 2D materials, consequently improving charge carrier mobility. The presence of ripples in

the MoS₂ induced by a bulged substrate brings about a change in the dielectric constant and suppresses photon scattering, resulting in enhanced mobility.

Also, a first-principles calculation has been performed to obtain this mechanism of monolayer MoS₂. The calculation details are as follows:

Our first-principles calculations were performed with Vienna Ab-initio Simulation Package (VASP)^{2, 3} using the projector augmented wave (PAW)⁴ method. The exchange-correlation interactions were handled with Perdew-Burke-Ernzerhof (PBE)⁵ functional. The cutoff energy for the plane-wave expansion was set to 400 eV in the whole process. The atom positions were relaxed until forces on them were less than 10⁻² eV/Å. The rippled MoS₂ was constructed by sin function based on the 1×3×1 supercell of orthogonal MoS₂. The rippled heights were selected 1, 2, and 3 Å, respectively, corresponding to the vertical distance between the highest and lowest Mo atoms. For the first Brillouin zone sampling, Γ -centered Monkhorst-Pack k-point meshes of 8×2×1 and 16×3×1 were used to perform structural relaxation and optical properties calculations. In addition, a vacuum layer larger than 15 Å was added to reduce the mirror interactions.

To elucidate the impact of substrate-induced bulges on the dielectric constant, we conducted dielectric function calculations for monolayer MoS₂ supercells employing four idealized models: flat MoS₂ and corrugated MoS₂ with curvature heights of 1 Å (Rippled-1 Å), 2 Å (Rippled-2 Å), and 3 Å (Rippled-3 Å). Progressing from flat MoS₂ to corrugated-1 Å, and from Rippled-1 Å to Rippled-3 Å, the relationship reveals an increasing curvature height, resulting in non-uniform strain within the twisted MoS₂. The curvature height is defined as the difference in height between the highest and lowest Mo atoms, as illustrated in Fig. R3.1a-d. Energy band structure calculations for these four model structures are presented in Fig. R3.1e-h, indicating a progressively diminishing energy band in MoS₂ as curvature increases.

To further investigate the relationship of the mobility of MoS₂, the mobility μ can be obtained by the following⁶:

$$\mu = \frac{M_M M_X A t^2 \varepsilon^2 (\hbar \omega)^2}{16 \pi^2 e^3 n m^* Z_{MB}^2 (\sqrt{M_M} + \sqrt{M_X})^2} \quad (1)$$

where the M_M , and M_X are the atomic mass of metal and chalcogen atoms, A is the area of the unit cell, n is the Bose–Einstein distribution, t is the effective thickness, ε is the

in-plane optical dielectric constant, Z_{MB} is the Born effective charge of the M. t and ϵ can be approximated by the bulk dielectric constant and the interlayer distance in the bulk material. In accordance with Equation (1), it is observed that the material's mobility is directly proportional to ϵ^2 . The calculations of the real part of the dielectric function (ϵ_1) for the aforementioned four model structures are presented in Fig. R3.2 and Fig. R3.3, with the static permittivity of MoS₂ determined by the value at $E \rightarrow 0$. It is noteworthy that these calculations exclude the ionic contribution, as its impact is negligible in comparison to the electronic contribution. The results clearly demonstrate that both the in-plane and out-of-plane dielectric constants of MoS₂ exhibit an increase with the height of curvature. Consequently, the ripples in the MoS₂ caused by the bulged substrate lead to a change in the dielectric constant, which leads to mobility enhancement.

Fig. R3.1 | 3D views and electronic band structures of rippled MoS₂ crystal from different curvature height. a-d, 3D views of (a) flat-MoS₂ and rippled-MoS₂ with (b) 1 Å, (c) 2 Å and (d) 3 Å curvature heights. e-h, Electronic band structures of (e) flat-MoS₂ and rippled-MoS₂ with (f) 1 Å, (g) 2 Å and (h) 3 Å curvature heights.

Fig. R3.2 | First-principles calculations of the dielectric function's real part (ϵ_1) for monolayer MoS₂. a-d, Flat-MoS₂ and rippled-MoS₂ with curvature heights of 1 Å, 2 Å, and 3 Å.

Fig. R3.3 | Comparison of the static dielectric constants of monolayers of MoS₂ with varying curvature heights (flat-rippled, rippled-1 Å, rippled-2 Å, and rippled-3 Å), calculated based on first principles. $\epsilon_{||}$ and ϵ_{\perp} represent in-plane and out-of-plane dielectric constants of MoS₂, respectively.

2. Experiment verification. To better evaluate the enhancement brought by the bulged interface, we've fabricated devices with flat, less rippled and rippled interfaces. By calculation, we get their mobility as $6.295 \text{ cm}^2 \text{ V}^{-1} \text{ s}^{-1}$, $26.73 \text{ cm}^2 \text{ V}^{-1} \text{ s}^{-1}$ and $97.77 \text{ cm}^2 \text{ V}^{-1} \text{ s}^{-1}$, respectively. The degradation is obvious as the roughness decreases (Fig. R3.4a-f). The introduction of the rippled interface as the dielectric layer results in a relative enhancement in mobility compared to devices on a flat SiN_x surface. Also, we've fabricated a device array of 18 by 18 over the bulged SiN_x surface and conducted a statistical analysis of the mobility (Fig. R3.5). The average mobility is $116.5 \text{ cm}^2 \text{ V}^{-1} \text{ s}^{-1}$ and the standard deviation is $81.289 \text{ cm}^2 \text{ V}^{-1} \text{ s}^{-1}$.

The implementation of all-day motion detection is based on the positive and negative photo response and the memory of the device. Due to high mobility, both the photocurrent and its dynamic range get larger (**Fig. R3.6a-c**). As the mobility declined for roughness decrement, both the electrical and optoelectrical performance decay.

Fig. R3.4 | Roughness and transfer curves of devices on both flat and rippled silicon nitride interface. a, Roughness of the device on the flat interface by AFM. b, Roughness of device on the rippled interface with less roughness by AFM. c, Roughness of device on the pristine rippled sr-SiN_x interface by AFM. d, The transfer curve and mobility of the device on the flat interface. The mobility is $6.295 \text{ cm}^2 \text{ V}^{-1} \text{ s}^{-1}$. e, The transfer curve and mobility of the device on the rippled interface with less roughness. The mobility is $26.73 \text{ cm}^2 \text{ V}^{-1} \text{ s}^{-1}$. f, The transfer curve and mobility of the device on the pristine rippled interface. The mobility is $97.77 \text{ cm}^2 \text{ V}^{-1} \text{ s}^{-1}$, which shows a remarkable enhancement brought by the rippled interface.

Fig. R3.5 | Electrical statistics of the RAO array. a, Transfer curves of 262 pixels with a high yield of 80% at the $V_{DS} = 0.5$ V. The deep blue curve represents a typical transfer curve of each RAO processor, which shows a large memory window. b, Mobility statistics of the RAO array illustrate a normal distribution. The maximum and average mobilities are 406.7 and $116.5 \text{ cm}^2\text{V}^{-1}\text{s}^{-1}$.

Fig. R3.6 | Comparison of the photocurrent and dynamic range of the phototransistors on different substrates. a, Surface of the specially treated sr-SiN_x measured by AFM with smaller roughness. b, Surface of the specially treated sr-SiN_x measured by AFM with normal roughness. c, Comparison of the photocurrent with different roughnesses. d, Dynamic range of the all-in-one device with different

roughness.

Revision:

We have added Section 2, Mechanism of mobility enhancement and first-principles calculation, Supplementary and **Supplementary Fig. 7-9** to show the details of the first-principles calculation of mobility enhancement (Page 32-34). Also, we've added **Supplementary Figure. 4** to display the comparison of devices over flat and rippled silicon nitride interfaces (Page 30). Corresponding discussions have been incorporated into the revised manuscript as follows: “*Besides, a first-principles calculation is performed to obtain this mobility enhancement of monolayer MoS₂, of which the detailed discussion is shown in Supplementary Section 2*”. (Page 6)

Reference:

1. Ng, H. K. et al. Improving carrier mobility in two-dimensional semiconductors with rippled materials. *Nat. Electron.* **5**, 489-496 (2022).
2. Kresse, G. & Furthmüller, J. Efficient iterative schemes for ab initio total-energy calculations using a plane-wave basis set. *Phys. Rev. B* **54**, 11169-11186 (1996).
3. Kresse, G. & Furthmüller, J. Efficiency of ab-initio total energy calculations for metals and semiconductors using a plane-wave basis set. *Comput. Mater. Sci.* **6**, 15-50 (1996).
4. Blöchl, P. E. Projector augmented-wave method. *Phys. Rev. B* **50**, 17953-17979 (1994).
5. Perdew, J. P., Burke, K. & Ernzerhof, M. Generalized Gradient Approximation Made Simple. *Phys. Rev. Lett.* **77**, 3865-3868 (1996).
6. Cheng, L. & Liu, Y. What Limits the Intrinsic Mobility of Electrons and Holes in Two Dimensional Metal Dichalcogenides? *J. Am. Chem. Soc.* **140**, 17895-17900 (2018).

Comments 2:

The other thing I would request the author to do is checking the mobility on control samples without ripples and provide averages and standard deviations.

Response:

Thank the reviewer for the valuable comment. To better evaluate the mobility enhancement brought by the ripples, we have decreased the roughness of the sr-SiN_x dielectric layer as the control sample and then made a device array scaled 8 by 8 over it to better analyze the mobility enhancement induced by the rippled interface

quantitatively. Bulged substrates with smaller roughness are obtained using a wet polishing method (85 %wt phosphoric acid, 170 °C) where bulges are smoothed with increasing etching time (shown in Fig. R3.7). The relationship between and device mobility is shown in **Fig. R3.4**. Microscope and SEM images of the MDR array on the control sample are shown in **Fig. R3.8**. Further, we conducted a statistical analysis of both the electrical properties and the mobility (**Fig. R3.9**). On the control sample, the average mobility of devices is $31.3 \text{ cm}^2 \text{ V}^{-1} \text{ s}^{-1}$ and the standard deviation of the mobility of devices is $18.84 \text{ cm}^2 \text{ V}^{-1} \text{ s}^{-1}$.

Fig. R3.7 | Morphology of bulged SiN_x/Si substrates: AFM images of SiN_x/Si rippled substrate morphology with different roughness obtained using the wet polishing method. a, pristine surface of bulged SiN_x/Si substrates, of which the standard deviation of the roughness is 3.407 nm. b, Surface with the process of wet polishing method (85 %wt phosphoric acid, 170 °C) for 3 minutes, of which the standard deviation of the roughness is 2.745 nm. c, Surface with the process of wet polishing method (85 %wt phosphoric acid, 170 °C) for 9 minutes, of which the standard deviation of the roughness is 1.760 nm.

Fig. R3.8 | Microscope and SEM images of the MDR array on the control sample. a, Microscope image of the array on the control sample. SEM image of the array on the control sample.

Fig. R3.9 | Electrical statistics of the RAO array over the surface with less roughness. a, Transfer curves of 57 pixels at the $V_{DS} = 0.5$ V. The deep blue curve represents a typical transfer curve of each RAO processor, which shows a large memory window same as the device over a pristine surface. b, Mobility statistics of the RAO array over the surface with less roughness, illustrating a normal distribution. The maximum and average mobilities are 78.5 and $31.3 \text{ cm}^2 \text{V}^{-1} \text{s}^{-1}$.

Revision:

We have added **Supplementary Fig. 5** to show the wet polishing method and the results during the polishing process (Page 30-31). Corresponding discussions have been incorporated into the revised manuscript as follows: *“The wet polishing method is used to decrease the roughness of the substrate, and a more detailed demonstration is shown in Supplementary Fig. 5”*. (Page 5)

Comments 3:

Finally, the mobility should also be checked as a function of channel length to isolate effect of the contact resistance vs channel resistance.

Response:

To isolate the effect of the contact resistance and the channel resistance, we employed the Transmission Line Model (TLM), aiming to determine the specific

contact resistivity of a metal-semiconductor junction. RAO devices were fabricated with varying channel lengths, ranging from 6 μm to 14 μm , over a control sample with lower roughness. The total resistance for each device was calculated, as depicted in **Fig. R3.10**. A linear fitting analysis was performed, shown as the dashed line in **Fig. R3.10**. The intercept of the linear fitting equals 2 times of contact resistance, as the slope equals the resistivity over the width of the metal pad. In this case, the mobility of this device using TLM is modulated from 26.76 $\text{cm}^2 \text{V}^{-1} \text{s}^{-1}$ to 34.57 $\text{cm}^2 \text{V}^{-1} \text{s}^{-1}$ according to the equation as follows: $\mu = \frac{L}{qnC_{\text{ox}}R_{\text{channel}}}$, where μ is the mobility, L is the channel length, q is the elementary charge (approximately 1.602×10^{-19} C), n is the electron concentration, C_{ox} is the oxide capacitance per unit area. R_{channel} is the intrinsic channel resistance.

Fig. R3.10 | Total resistance of channel and contact and its linear fitting against channel length. The total resistance of the channel and contact increases as the channel length increases. The intercept is 26 k Ω , two times the contact resistance, which means the contact resistance is 13 k Ω .

Revision:

We have added Section 11, Contact resistance and its influence on mobility calculation, Supplementary and **Supplementary Fig. 32** to illustrate the influence brought by the contact resistance (Supplementary Page 56). Corresponding discussions have been incorporated into the revised manuscript as follows: *“To discern the impact of contact resistance and channel resistance, we utilized a Transmission Line Model to determine the specific contact resistivity of the metal-semiconductor junction (shown in Supplementary Section 11).”*. (Page 9)

Comments 4:

The photo detection and motion detection data are great, I don't have much comments about it. I only want the authors to be clear and comment if the image capture and motion detection is done by the whole 2D array or just by a single pixel and then rest is simulation. It seems to me that the pixels are not individually wired to sense. So the perception of image and motion capture appears misleading.

Response:

We thank the reviewer for the valuable comment. The image is captured by a traditional CMOS sensor and the motion detection is done based on data from the whole 2D array (18 by 18) by simulation on MATLAB. The simulation of motion detection is implemented based on inter-frame differential computation in the following process: first, we experimentally record the negative photocurrent and the positive photocurrent of each device in an array scaled 18 by 18, and then transfer them to weights in the mapping matrix. Next, we set a suitable time interval Δt based on the motion pattern of the trolleys and extract the two frames ($t_1, t_1+\Delta t$) at the interval Δt , as shown in Fig. R3.11. The prior frame pixels are mapped by the positive mapping matrix and stored, due to the non-volatile photoconductive memory properties. The latter frame pixels are mapped by the negative mapping matrix and the memorized result is combined with the mapping result of the previous frame. After obtaining the inter-frame sum, a threshold is defined to help differentiate the summed data. Finally, we reconstruct the classified data and transform it into a detected image.

Fig. R3.11 | Process for all-day motion detection and recognition. The all-day MDR methodology using inter-frame differential computation. The array of positive photoconductivity and negative photoconductivity multiply the frame n and frame n+1 with a constant interval Δt , respectively. The sum of the two array makes the static part of the frame cancels off, which allows for motion detection.

Revision:

We have revised Section 12, Methodology for motion detection and recognition, Supplementary and added a thorough process of motion detection. (Page 57).

Responses to Reviewer # 4:

In this work, the authors report some very interesting results on the smart sensing/imaging by combining spectral sensing, storage and simple computation functions all on chip. In my view, optical sensing and imaging are at the verge of substantial transformation. This work may represent one of such changes. Overall, the results are interesting and sound. I think it should be published in Nature Communications after they address the following issue.

Response:

We thank the reviewer for the positive comments “*the results are interesting and sound*”. In response to the concerns raised by the reviewer regarding the distinction between regular devices without ripples and RAO devices, as well as the mechanisms contributing to performance improvement in RAO devices, we conducted a comprehensive assessment. This involved providing an in-depth analysis of the characteristics of the devices over surfaces with varying roughness. Additionally, we elucidated the underlying mechanisms responsible for the observed differences. With the help of the reviewer, the whole manuscript has been largely improved. In the following, we will address all comments point-by-point and revised the manuscript. We hope that the revised manuscript would remove the reviewers’ concerns.

Comments:

It seems that this so-called "Ripple assisted optoelectronic (ROC)" devices play an important role. I suggest that the authors clarify its role. For example, will this demonstration still valid if regular devices with no ripples are utilized? If yes, what is expected performance degradation with regular devices. Also what are the mechanisms for the performance improvement in ROC? All these issues should be thoroughly discussed.

Response:

We thank the reviewer for the valuable comment. In the following context, we’ll make a thorough assessment to clarify the role of the ripple assisted optoelectronic (RAO) device from two aspects providing the analysis of characteristics of the devices over surfaces with different roughness, and the mechanism that brought to this

difference.

1. Electrical and optoelectrical properties over different surfaces

Firstly, we have provided both electrical and optoelectrical properties over different surfaces of pristine rough surface, less rough surface and flat surface after the introduction to the method for smoothing the substrates.

Bulged substrates with smaller roughness are obtained using a wet polishing method (85 %wt phosphoric acid, 170 °C) where bulges are smoothed with increasing etching time (shown in **Fig. R4.1**). The relationship between and device mobility is shown in Fig. R4.2.

Fig. R4.1 | Morphology of bulged SiN_x/Si substrates: AFM images of SiN_x/Si rippled substrate morphology with different roughness obtained using the wet polishing method. a, pristine surface of bulged SiN_x/Si substrates, of which the standard deviation of the roughness is 3.407 nm. b, Surface with the process of wet polishing method (85 %wt phosphoric acid, 170 °C) for 3 minutes, of which the standard deviation of the roughness is 2.745 nm. c, Surface with the process of wet polishing method (85 %wt phosphoric acid, 170 °C) for 9 minutes, of which the standard deviation of the roughness is 1.760 nm.

1.1. Mobility over different roughness

To better evaluate the enhancement brought by the bulged interface, we've fabricated devices with flat, less rippled and rippled interfaces. The degradation is obvious as the roughness decreases (**Fig. R4.2a-c**).

Further, to ensure the fidelity of the mobility calculations, we conducted a statistical analysis of the mobility of two device arrays of rippled and less rippled. The mobility distribution is shown in **Fig. R4.3a-b**. For rippled interface, the average mobility is $116.5 \text{ cm}^2 \text{ V}^{-1} \text{ s}^{-1}$, while for the devices over the interface with less roughness,

the average mobility is $31.3 \text{ cm}^2 \text{ V}^{-1} \text{ s}^{-1}$, which is smaller than the devices over pristine sr-SiN_x dielectric. This strongly proves that our device has remarkable mobility enhancement characteristics.

Fig. R4.2 | Roughness and transfer curves of devices on both flat and rippled silicon nitride interface. a, Roughness of the device on the flat interface by AFM. b, Roughness of device on the rippled interface with less roughness by AFM. c, Roughness of device on the pristine rippled sr-SiN_x interface by AFM. d, The transfer curve and mobility of the device on the flat interface. The mobility is $6.295 \text{ cm}^2 \text{ V}^{-1} \text{ s}^{-1}$. e, The transfer curve and mobility of the device on the rippled interface with less roughness. The mobility is $26.73 \text{ cm}^2 \text{ V}^{-1} \text{ s}^{-1}$. f, The transfer curve and mobility of the device on the pristine rippled interface. The mobility is $97.77 \text{ cm}^2 \text{ V}^{-1} \text{ s}^{-1}$, which shows a remarkable enhancement brought by the rippled interface.

Fig. R4.3 | Mobility distribution of device array with dielectric of different interface roughness. a, Mobility distribution of device array with pristine sr-SiN_x and the Gauss fit of the distribution. The maximum and average mobilities are 406.7 and 116.5 cm² V⁻¹ s⁻¹. b, Mobility statistics of the array with less roughness dielectric and the Gauss fit of the distribution. The maximum and average mobilities are 78.5 and 31.3 cm² V⁻¹ s⁻¹.

1.2. Degradation of optoelectrical properties of the device over the less rough surface.

Fig. R4.4 shows the comparison of the photocurrent and dynamic range of the phototransistors on different substrates. Here, we define the difference between the maximum and minimum currents as the optical storage dynamic range of the all-in-one device. It is worth noting that the Dynamic Range of the device fabricated on the smoother surface is much smaller than that on the rougher counterpart.

Fig. R4.4 | Comparison of the photocurrent and dynamic range of the phototransistors on different substrates. a, Surface of the specially treated sr-SiN_x measured by AFM with smaller roughness. b, Surface of the specially treated sr-SiN_x measured by AFM with normal roughness. c, Comparison of the photocurrent with different roughnesses. d, Dynamic range of the all-in-one device with different roughnesses.

2. Mechanism that brought this difference

2.1. Mobility enhancement

In the Nature electronics paper¹, lattice distortions can reduce electron–phonon scattering in 2D materials and thus improve the charge carrier mobility. The ripples in the MoS₂ caused by the bulged substrate lead to a change in the dielectric constant and a suppressed photon scattering, which leads to mobility enhancement.

Also, a first-principles calculation has been performed to obtain this mechanism of monolayer MoS₂. The calculation details are as follows:

Our first-principles calculations were performed with Vienna Ab-initio Simulation Package (VASP)^{2, 3} using the projector augmented wave (PAW)⁴ method. The exchange–correlation interactions were handled with Perdew–Burke–Ernzerhof (PBE)⁵ functional. The cutoff energy for the plane-wave expansion was set to 400 eV in the

whole process. The atom positions were relaxed until forces on them were less than 10^{-2} eV/Å. The rippled MoS₂ was constructed by sin function based on the $1 \times 3 \times 1$ supercell of orthogonal MoS₂. The rippled heights were selected 1, 2, and 3 Å, respectively, corresponding to the vertical distance between the highest and lowest Mo atoms. For the first Brillouin zone sampling, Γ -centered Monkhorst-Pack k-point meshes of $8 \times 2 \times 1$ and $16 \times 3 \times 1$ were used to perform structural relaxation and optical properties calculations. In addition, a vacuum layer larger than 15 Å was added to reduce the mirror interactions.

To elucidate the impact of substrate-induced bulges on the dielectric constant, we conducted dielectric function calculations for monolayer MoS₂ supercells employing four idealized models: flat MoS₂ and corrugated MoS₂ with curvature heights of 1 Å (Rippled-1 Å), 2 Å (Rippled-2 Å), and 3 Å (Rippled-3 Å). Progressing from flat MoS₂ to corrugated-1 Å, and from Rippled-1 Å to Rippled-3 Å, the relationship reveals an increasing curvature height, resulting in non-uniform strain within the twisted MoS₂. The curvature height is defined as the difference in height between the highest and lowest Mo atoms, as illustrated in Fig. R4.5a-d. Energy band structure calculations for these four model structures are presented in Fig. R4.5e-h, indicating a progressively diminishing energy band in MoS₂ as curvature increases.

To further investigate the relationship of the mobility of MoS₂, the mobility μ can be obtained by the following⁶:

$$\mu = \frac{M_M M_X A t^2 \varepsilon^2 (\hbar \omega)^2}{16 \pi^2 e^3 n m^* Z_{MB}^2 (\sqrt{M_M} + \sqrt{M_X})^2} \quad (1)$$

where the M_M , and M_X are the atomic mass of metal and chalcogen atoms, A is the area of the unit cell, n is the Bose–Einstein distribution, t is the effective thickness, ε is the in-plane optical dielectric constant, Z_{MB} is the Born effective charge of the M. t and ε can be approximated by the bulk dielectric constant and the interlayer distance in the bulk material. In accordance with Equation (1), it is observed that the material's mobility is directly proportional to ε^2 . The calculations of the real part of the dielectric function (ε_1) for the aforementioned four model structures are presented in **Fig. R4.6** and **Fig. R4.7**, where the static permittivity of MoS₂ is determined by the value at $E \rightarrow 0$. It is noteworthy that these calculations exclude the ionic contribution, as its impact

is negligible in comparison to the electronic contribution. The results demonstrate that both the in-plane and out-of-plane dielectric constants of MoS₂ exhibit an increase with the height of curvature. Consequently, the ripples in the MoS₂ caused by the bulged substrate lead to a change in the dielectric constant, which leads to mobility enhancement.

Fig. R4.5 | 3D views and electronic band structures of rippled MoS₂ crystal from different curvature height. a-d, 3D views of (a) flat-MoS₂ and rippled-MoS₂ with (b) 1 Å, (c) 2 Å and (d) 3 Å curvature heights. e-h, Electronic band structures of (e) flat-MoS₂ and rippled-MoS₂ with (f) 1 Å, (g) 2 Å and (h) 3 Å curvature heights.

Fig. R4.6 | First-principles calculations of the dielectric function's real part (ϵ_1) for monolayer MoS₂. a-d, Flat-MoS₂ and rippled-MoS₂ with curvature heights of 1 Å, 2 Å, and 3 Å.

Fig. R4.7 | Comparison of the static dielectric constants of monolayers of MoS₂ with varying curvature heights (flat-rippled, rippled-1 Å, rippled-2 Å, and rippled-3 Å), calculated based on first principles. ϵ_{\parallel} and ϵ_{\perp} represent in-plane and out-of-plane dielectric constants of MoS₂, respectively.

2.2. Optoelectrical enhancement

The silicon-rich SiN_x dielectric layer contains numerous natural defects that are capable of trapping holes. It is worth mentioning that heavily p-doped silicon functions as a natural hole reservoir and can provide a sufficient number of holes for injection. During the writing process, a +20 V gate voltage is applied to drive hole injection from the heavily p-doped silicon to the sr- SiN_x dielectric. The injected holes are subsequently trapped by hole trapping centers, which act as positive local gates that enhance the electron concentration and channel conductance, thereby corresponding to the memory of the low resistance state, which is shown in **Fig. R4.8a-b**. For the light erase process, a fixed -3 V gate voltage is applied accompanied by the optical stimulus, in which the holes in the dielectric are released. The reduction of stored holes leads to a smaller channel current, resulting in negative photo conductance (NPC). The carrier distribution and band diagram of NPC are shown in **Fig. R4.8c**.

During the erasing process, a fixed -20 V gate voltage is applied to release the trapped holes, leading to a decrease in channel conductance that corresponds to the memory of the high resistance state (shown in **Fig. R4.8d-e**). As for the light write process, a fixed 3 V gate voltage is applied accompanied by the optical stimulus, in which the holes in the dielectric are trapped as the photon carriers generated by optical stimuli overcome the $\text{MoS}_2/\text{sr-SiN}_x$ interface barrier (shown in **Fig. R4.8f**), leading to the positive photo conductance (PPC).

Therefore, as the electron concentration of the channel is tuned by the injection or release of holes in the dielectric layer, the channel conductance is dependent on the carrier mobility if other conditions remain unchanged. As the mobility enhancement brought by the ripples of the bulged substrate, the responsivity gets larger as well and the photocurrent gets larger, leading to the increment of the dynamic range of the RAO device.

Fig. R4.8 | Positive and negative photoconductive mechanisms. a-b, Writing process causes the device to transition to a low-resistance state. **c,** Light erase process and negative photo conductance. **d-e,** Erasing process makes the device return to a high resistance state. **f,** Light write process and positive photo conductance.

Overall, thank you again for your insightful comments. We appreciate your constructive suggestions and have carefully reevaluated our study in light of your suggestions. The manuscript has been revised to address the concerns raised, and a thorough assessment of the mobility enhancement and the unique properties it brings has been made. We have also taken steps to provide more solid theory and evidence. We believe these revisions enhance the overall quality and credibility of our work. Your feedback has been invaluable, and we are grateful for the opportunity to improve our manuscript based on your recommendations.

Revision:

We have added Section 2, Mechanism of mobility enhancement and first-principles calculation, Supplementary and **Supplementary Fig. 7-9** to show the details of the first-principles calculation of mobility enhancement (Page 32-34). Also, we've added **Supplementary Figure. 4** to display the comparison of devices over flat and rippled silicon nitride interfaces (Page 30). Besides, we have revised Section 3, Memory

and PPC, NPC mechanisms of the RAO processor, Supplementary and added **Supplementary Fig. 12** to better illustrate the mechanisms of PPC and NPC. (Page 37-38) Corresponding discussions have been incorporated into the revised manuscript as follows: “*The wet polishing method is used to decrease the roughness of the substrate, and a more detailed demonstration is shown in **Supplementary Fig. 5***” (Page 5). “*Besides, a first-principles calculation is performed to obtain this mobility enhancement of monolayer MoS₂, of which the detailed discussion is shown in **Supplementary Section 2***” (Page 5). “*For the light erase process, a fixed -3 V gate voltage is applied accompanied by the optical stimulus, in which the holes in the dielectric are released. The reduction of stored holes leads to a smaller channel current, resulting in negative photo conductance (NPC). The carrier distribution and band diagram of NV-PPC and NV-NPC are shown in **Supplementary Fig. 12***” (Page 5).

Reference:

1. Ng, H. K. et al. Improving carrier mobility in two-dimensional semiconductors with rippled materials. *Nat. Electron.* **5**, 489-496 (2022).
2. Kresse, G. & Furthmüller, J. Efficient iterative schemes for ab initio total-energy calculations using a plane-wave basis set. *Phys. Rev. B* **54**, 11169-11186 (1996).
3. Kresse, G. & Furthmüller, J. Efficiency of ab-initio total energy calculations for metals and semiconductors using a plane-wave basis set. *Comput. Mater. Sci.* **6**, 15-50 (1996).
4. Blöchl, P. E. Projector augmented-wave method. *Phys. Rev. B* **50**, 17953-17979 (1994).
5. Perdew, J. P., Burke, K. & Ernzerhof, M. Generalized Gradient Approximation Made Simple. *Phys. Rev. Lett.* **77**, 3865-3868 (1996).
6. Cheng, L. & Liu, Y. What Limits the Intrinsic Mobility of Electrons and Holes in Two Dimensional Metal Dichalcogenides? *J. Am. Chem. Soc.* **140**, 17895-17900 (2018).

Based on the reviewer's comments, we have performed the following correction:

Revised manuscript

1. Page 2, and 3, we have revised the abstract and introduction to temper any hyperbole and improve the readability and credibility of this paper.
2. Page 5, we have supplemented memory stability and the explanation including PPC and NPC, and added a discussion of them.
3. Page 6, we have supplemented a first-principles calculation to better illustrate the influence brought by the bulged substrate and added a description of enhancement. Besides, we have added a discussion of the photocurrent with varying light lasting time and supplemented the photo response under this varying condition.
4. Page 7 and 8, we have supplemented the generality and flexibility of sr-SiNx dielectric for its function provided and added the discussion of MoTe₂ devices on this substrate. Besides, we have discussed MoS₂ devices with both ME and CVD films and supplemented their consistency in both electrical and optoelectrical properties.
5. Page 9, we have revised the unit and description of the maximum and minimum current of each device of the RAO array.
6. Page 11, we have revised the conclusion to improve the readability and make a more reasonable summary.
7. Page 14, we have revised **Figure 2b** by adding two arrows showing the direction of transfer curves and revised the figure captions.
8. Page 16, we have revised Figure 3e, f, h. We revised the unit of **Figure 3a, e, f** to make it more reasonable. Besides, we have added more representative papers in **Figure 3h** to make the comparison more solid and sound.
9. Page 22, we have revised device fabrication. we have added the synthesis process of the CVD monolayer MoS₂.

Revised Supplementary Information

1. Page 30, we have revised **Supplementary Fig. 4** to add the mobility over a less rough substrate and revised the figure caption.
2. Pages 32 to 34, we have added **Section 2, Mechanism of mobility enhancement and first-principles calculation**, to explain the mechanism brought to the mobility enhancement, and have added **Supplementary Fig. 7-9** to display the results.
3. Pages 35 and 36, we have added **Section 3, Response time and responsivity demonstration**, to display the photo response with varying light lasting time and wavelength, and have added **Supplementary Fig. 10, 11** to show the results.
4. Pages 37 to 39, we have reorganized **Section 4, Memory and PPC, NPC mechanisms of the RAO processor**, and added a more solid and detailed explanation of NPC and PPC, and their retention characteristics of them. We have added **Supplementary Fig. 12-14** to display the results.
5. Pages 47 to 49, we have added **Section 9, Consistency between devices with mechanically exfoliated films and CVD films**, to illustrate the both electrical and optoelectrical consistency between devices with ME and CVD films, and have added **Supplementary Fig. 22-24** to display the results.
6. Pages 50 to 51, we have reorganized **Section 10, Generality and flexibility of sr-SiN_x**, to demonstrate the generality and flexibility of this sr-SiN_x dielectric by providing data that MoTe₂ devices exhibited similar features as MoS₂ devices. We have added **Supplementary Fig. 25-26** to show the results.
7. Page 56, we have reorganized **Section 11, Contact resistance and its influence on mobility calculation**, to discern the impact of contact resistance and channel resistance.
8. Page 57, we have reorganized **Section 12, Methodology for motion detection and recognition**, adding the detailed process of motion detection based on our RAO device array.

REVIEWERS' COMMENTS

Reviewer #1 (Remarks to the Author):

In the revised manuscript, the authors have responded the questions from the reviewers in a satisfactory manner. I would recommend the acceptance in the present form.

Reviewer #3 (Remarks to the Author):

The authors have addressed all my comments and have comprehensively revised the manuscript. It seems they have also addressed comments of the other reviewers. In my view this manuscript can be accepted at this point.

Reviewer #4 (Remarks to the Author):

In the revised version, the authors extensively addressed all the reviewers' comments and now it is ready for publication.